# In Praise of Stubbornness: An Empirical Case for Cognitive-Dissonance Aware Continual Update of Knowledge in LLMs

## Abstract

Through systematic empirical investigation, we uncover a fundamental property of large language models (LLMs) with implications for continual learning: they can safely learn facts that do not contradict existing knowledge, but attempts to update them with counterfactuals cause catastrophic corruption of *unrelated* knowledge. Unlike humans, who naturally resist conflicting information, LLMs have no such safeguards by design. This leads to severe interference, destroying up to 80% of unrelated factual knowledge even for as few as 10–100 counterfactual updates. To test whether selective plasticity can mitigate this damage, we perform targeted updates, distinguishing between previously used (*stubborn*) and rarely used (*plastic*) neurons. We find again an asymmetry: sparing frequently used neurons improves retention for non-contradictory updates (98% retained vs. 93% under standard updates), yet counterfactual updates trigger catastrophic interference regardless of targeting. This effect, which persists across tested models and scales (from GPT-2 to GPT-J-6B, as well as GPT-4.1, Llama3-8B, and Mistral-7B) and extends to diverse dataset types beyond synthetic counterfacts—with damage scaling with the degree of contradiction—suggests a general property of current LLMs. Finally, we show that counterfactual inputs can be detected with $\geq 95\%$ accuracy using simple model features, pointing to a practical safeguard. These findings motivate research on architectures that, like humans, naturally resist contradictions rather than allowing destructive overwrites.

## 1 Introduction

Humans protect their knowledge by *detecting* and *resisting* contradictions. When a new statement conflicts with what we already believe, we experience a psychological discomfort known as cognitive dissonance: the uncomfortable state of holding two conflicting beliefs simultaneously (Festinger, 1957; Croyle & Cooper, 1983). This discomfort drives us to hesitate, seek additional evidence and resolve the conflict through critical evaluation rather than passive acceptance. As a result, we often maintain both versions with appropriate episodic context ("Pluto was once classified a planet; today it is not") rather than simply overwriting existing knowledge (Van Veen et al., 2009). This cognitive *"stubbornness"* serves as a protective mechanism for knowledge integrity.

Large Language Models (LLMs) have, by design, no such contradiction filter. During gradient-based training, every sample, whether consistent or contradictory, updates the same weight space indiscriminately. This difference between human and artificial cognition led us to investigate a crucial question: *what happens when LLMs encounter contradictory information?* Through carefully controlled experiments across different model scales , we systematically compare how LLMs handle two types of knowledge updates: adding entirely new facts versus updating existing knowledge with counterfacts (e.g., training that "Paris is the capital of Italy" when the model knows "Paris is the capital of France"). Our investigation reveals, for the first time, a striking and concerning property: while LLMs tend to safely learn new, non-conflicting information, attempting to update them with counterfacts triggers catastrophic corruption of completely *unrelated* knowledge (Sec. 3). Even minimal contradictory updates (as few as 10-100 facts) can destroy up to 80% of a model's unrelated knowledge ( Fig. 4). Importantly, this effect persists across model scales and training approaches, and generalizes to other model families (GPT-4.1, Llama3-8B, Mistral-7B) and to diverse dataset

types including misinformation and insecure code—with damage correlating with the degree of contradiction (App. A, B). This suggests a fundamental limitation in how neural networks react to contradictions.

Next, the brain offers a second clue: it balances rigid and still-malleable circuits. For example, classic critical-period experiments showed that, once ocular-dominance columns are set in primary visual cortex, additional learning is minimal (Wiesel & Hubel, 1963). An analogous "use-it-early or lose-it" window shapes auditory-cortex tonotopy (Kral, 2013). Furthermore, the dentate gyrus keeps adding young plastic granule cells that excel at novel pattern separation, whereas older, less-plastic granules specialize for pattern completion, i.e., the recall of established memories (Clelland et al., 2009; Nakashiba et al., 2012). This coexistence of frozen and adaptable subnetworks points to selective plasticity as another potentially protective mechanism in biological systems, complementing cognitive dissonance. This leads us to investigate a second question: *could selective plasticity help artificial systems maintain knowledge integrity? And might the impact of such targeted updates differ also between contradictory and non-contradictory information?*

To investigate, we implemented an analogue of selective plasticity in LLMs by identifying historically over-used neurons, encoding old knowledge (which we make become "stubborn" in future updates) versus rarely used ones (which we leave "plastic" for future training). This led to our *second key discovery*: an intriguing asymmetry in how selective plasticity behaves with different types of updates (Sec. 4). While selective avoidance of stubborn neurons significantly improves old knowledge retention when adding new information (98% vs 93% with standard updates), counterfactual updates trigger catastrophic interference regardless of targeting strategy. This asymmetry reveals that while selective plasticity successfully protects knowledge during non-contradictory updates, it fails when applied to counterfactual overwrites, an operation that biological systems may avoid altogether. This finding points to a fundamental question: should we be trying to erase and replace conflicting knowledge, or preserve both versions with appropriate episodic context, ultimately turning all updates into non-conflicting ones? (see Implications)

Despite these catastrophic effects, conflicts are today a blind spot across LLM's continual learning and editing research, as briefly illustrated in Tab. 1 (see App. D for details). Warranting further investigation, this might also impact deployment as current post-training procedures lack any mechanism to identify conflicts before they are ingested. This motivated our last empirical investigation: *can we detect when information might be contradictory before training?* Through controlled experiments, we demonstrate that simple classifiers using (either internal or output) model features achieve 95%+ accuracy in distinguishing novel, familiar, and contradictory facts (Sec. 5, App.F.5) , offering hope for protective mechanisms.

Table 1: *Conflict awareness gap.* While continual learning explores many dimensions (memory usage, selective plasticity, etc.), conflict detection remains universally absent. Model editing focuses exclusively on contradictory updates by design, making it orthogonal to our challenge of continuous mixed-content integration that deployed LLMs require. See Appendix.D for an extended version.

| Examples | Incremental Type | Memory Usage | Task Awareness | Weight Plasticity | Architecture | Update Mechanism | Conflict Detection |
|---|---|---|---|---|---|---|---|
| iCaRL (Rebuffi et al., 2017) | Class-incremental | Replay | Task-Agnostic | Fixed | Fixed | Rehearsal | No |
| EWC (Kirkpatrick et al., 2017) | Task-incremental | None | Task-Aware | Selective | Fixed | Regularization | No |
| Progressive Nets (Rusu et al., 2016) | Task-incremental | None | Task-Aware | Fixed | Expanding | New Subnetworks | No |
| DEN (Yoon et al., 2017) | Task-incremental | None | Task-Aware | Selective | Expanding | Selective Expansion | No |
| GEM (Lopez-Paz & Ranzato, 2017) | Task-incremental | Replay | Task-Aware | Constrained | Fixed | Constrained Optimization | No |
| OWM (Zeng et al., 2019) | Task-incremental | None | Task-Aware | Orthogonal | Fixed | Orthogonal Projection | No |
| PackNet (Mallya & Lazebnik, 2018) | Task-incremental | None | Task-Aware | Selective | Fixed | Weight Masking | No |
| HAT (Serra et al., 2018) | Task-incremental | None | Task-Aware | Selective | Fixed | Attention Masking | No |
| ROME (De Cao et al., 2021) | Fact-incremental | None | Fact-Aware | Localized | Fixed | Rank-One Update | **N/A** |

**Implications and future work** Summarizing, our human-inspired empirical findings uncover for the first time a stark difference between non-dissonant updates (which LLMs handle robustly) and counterfacts (which trigger catastrophic corruption of totally unrelated knowledge). They also point to concrete opportunities such as the feasibility of dissonance detection and the benefits of selective plasticity for non-contradictory updates. Most critically, they show that the most straightforward historical approach (simply erasing and replacing old knowledge) leads to catastrophic forgetting of *unrelated* information. This contrasts sharply with human cognition, where we maintain both old and new knowledge with appropriate *episodic* (temporal) context. Consider again how humans handled learning that Pluto was no longer classified as a planet: rather than completely erasing our

previous understanding, we maintained both pieces of knowledge, understanding their historical context ("Pluto *was* once classified as a planet; *today* it is not").

Our discoveries inspire an intriguing hypothesis which deserves exploration in future work: perhaps the brain's remarkable stability emerges from three complementary mechanisms: (i) cognitive dissonance for detecting and resolving contradictions, (ii) append-only updates that preserve both old and new knowledge with appropriate episodic context (avoiding direct overwrites), and (iii) selective plasticity for protected storage. While the exact nature of biological knowledge integration remains an active area of research, our empirical findings motivate exploring fundamentally different artificial architectures—ones that might incorporate the above protective mechanisms rather than attempting to blindly overwrite existing knowledge.[1]

## 2 EXPERIMENTAL PIPELINE

Our empirical investigation comprises two main components: (i) a knowledge update pipeline to compare dissonant vs. non-dissonant updates with selective plasticity strategies (Fig. 1), and (ii) a classification pipeline to detect dissonant information using model features (Sec. 2.3).

As illustrated in Fig. 1, the update pipeline proceeds in two phases. In the first, we ensure the model learns a set of baseline "old" facts, while simultaneously collecting gradient and activation statistics to identify *stubborn neurons*, i.e. those heavily involved in encoding this baseline knowledge (detailed in Sec. 2.2.1). In the second, we learn the new facts (non-dissonant) followed by the counterfacts of the latter (dissonant). Before performing any fine-tuning on the new facts, we run a simulated forward/backward pass (without weight updates) to identify *candidate* and *specific* neurons preferred for storing these new facts (Sec. 4). We then systematically experiment with different neuron targeting strategies (Plastic, Stubborn, Candidate, Specific) to understand how knowledge placement affects retention of unrelated information.

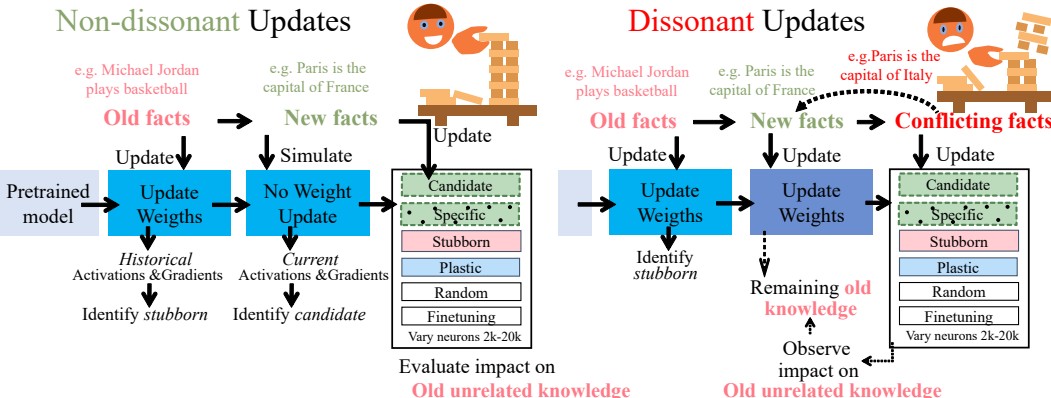

Figure 1: *Knowledge Update Pipeline*. In each experiment, we first establish baseline knowledge while tracking neuron usage to identify "stubborn" neurons. We then identify preferred neurons for new facts via a simulated pass, then tests different placement strategies (Plastic, Stubborn, Candidate, Specific; defined in Sec. 4). See main text for detailed walkthrough and Sec. 2.1 for protocol details.

### 2.1 USING COUNTERFACT DATASET TO CREATE DISSONANT AND NON-DISSONANT UPDATES

We design our controlled experiments using the COUNTERFACT dataset (Meng et al., 2022a), which contains approximately 17,000 factual statements along with their corresponding counterfacts, enabling controlled investigation of both non-contradictory knowledge integration and contradictory updates.

**Definition of Update Types.** We distinguish between:

---

[1]Anonymized code available at `https://figshare.com/s/81f7108d823b5e08e8ec`

- *Non-dissonant updates*: Adding entirely new facts that do not contradict existing knowledge (e.g., learning "Madrid is the capital of Spain" when the model does not know it).

- *Dissonant updates*: Attempting to modify existing knowledge with contradictory information (e.g., training that "Paris is the capital of Italy" when the model knows "Paris is the capital of France").

**Experimental Protocol.** Our investigation starts by establishing a baseline.

*Old "unrelated" knowledge:* we train models on 2,000 initial facts from COUNTERFACT until reaching near 100% accuracy. From this common baseline, we investigate two distinct update scenarios:

*Non-dissonant Updates:* We introduce 1,000 new facts (different general knowledge from COUNTERFACT) and train until convergence. *Note that these facts are non-dissonant to both the model's pre-training knowledge and recently learned facts.* We verify learning of new facts and measure retention of old baseline facts, revealing the model's capability for safe knowledge integration.

*Dissonant Updates:* After first learning 1,000 new facts (as above), we extract the subset of original facts still remembered accurately. We then introduce counterfacts that directly contradict the recently learned 1,000 facts (we also conduct experiments with smaller contradiction sets of 10 and 100). *These counterfacts conflict with both the model's pre-training knowledge and recently learned facts.* This "doubly dissonant" nature arises because COUNTERFACT's facts are general knowledge that pre-trained models were most likely exposed to during training. For instance, despite its small size, GPT-2-small already knows approximately 600 of them before any fine-tuning. We measure both learning of these new contradictory statements and retention of the remaining unrelated facts identified earlier. Finally, control experiments with a third round of non-dissonant updates, instead of dissonant, confirm the catastrophic effect is due to contradictions, not repeated updates.

**Evaluation Protocol.** For all stages, unless explicitly stated otherwise, we intentionally select hyperparameters (an example in App. G.2) that ensure successful learning of the target facts. This controlled setting allows us to isolate our key question: when models successfully learn new information, how much damage occurs to unrelated knowledge? Knowledge retention is measured at the end of each training stage after convergence.

**Cross-validation.** We employ 5-fold cross-validation by creating different splits of the 2,000 initial and 1,000 new facts from COUNTERFACT's 17K facts. This ensures our findings are robust across different subsets of general knowledge facts rather than specific to particular fact selections. Notably, we do not need to explicitly control for facts being "unknown" at pre-training, as COUNTERFACT's general knowledge nature means models have likely seen similar information during pre-training—our training simply helps them achieve reliable accuracy on these facts, and ease our measurements.

**Models and Implementation.** We employ models from the GPT family to enable comparison across scales, focusing on GPT-2-small and GPT-2-xl, with additional validation on GPT-J-6B. All experiments use 5-fold cross-validation, varying the specific sets of old and new facts. Implementation uses Hugging Face Transformers on NVIDIA GPUs.

Importantly, the small dataset size (17,000 facts) creates different visibility into catastrophic forgetting across model scales. Effects are most clearly observed with GPT-2-small, where our 2000 baseline facts at each fold represent a larger portion of the model's knowledge. In GPT-2-xl, the same number of tracked facts represents a smaller fraction of total knowledge, leading to less visible forgetting in non-dissonant cases. We therefore focus our main text on GPT-2-small results while providing aligned GPT-2-xl findings in the Appendix.

**Evaluation Metrics.** We measure performance using the standard factual accuracy metric from the model editing literature: the percentage of facts correctly recalled. This allows us to track both new knowledge acquisition and old knowledge retention. As revealed in Fig. 4, plotting these two dimensions against each other exposes the stark asymmetry between dissonant and non-dissonant updates, our first key finding.

## 2.2 EXPERIMENTING WITH SELECTIVE PLASTICITY

To implement selective plasticity, we follow the experimental pipeline illustrated in Fig. 1, systematically exploring where to selectively place new knowledge based on historical neuron usage.

### 2.2.1 Historical tracking during baseline knowledge learning

During the initial training on the 2,000 baseline "unrelated knowledge" facts, we maintain an aggregate profile of neuronal activity by accumulating activations and gradients at every training step. For each output dimension $n$ in the Transformer blocks, including feed-forward (MLP) layers and attention projections (K, Q, V matrices), we record:

- $A_n(t)$: the activation (layer output) for dimension $n$ at training step $t$
- $G_n(t)$: the gradient of the loss with respect to that output at step $t$

We then compute cumulative historical measures by summing over $T$ training steps:

- $H\hat{G}_n = \sum_{t=1}^{T} G_n(t)$: the cumulative *historical gradient* (signed values)
- $H\hat{A}_n = \sum_{t=1}^{T} A_n(t)$: the cumulative *historical activation*

To mitigate scale differences across layers, we optionally standardize gradients per layer before accumulation. The full derivation, including batch aggregation and token-level processing, is provided in Appendix E. We use the gradient[2] historical activity to classify neurons as plastic" or stubborn" based on their past usage during baseline learning, as follows, and as visually illustrated in Fig.2.

**Plastic Neurons.** Neurons underutilized during baseline learning. To identify them, we rank neurons by increasing historical gradient values and select the top $N$ neurons with the lowest cumulative gradients:

$$\mathcal{N}_{\text{plastic}} = \{n \mid \text{rank}(H\hat{G}_n) \leq N\},$$

where $H\hat{G}_n$ is the historical gradient for neuron $n$. This allows to assess whether targeting underutilized neurons can integrate new knowledge without interfering with baseline knowledge.

**Stubborn Neurons.** Neurons that accumulated high historical gradients during baseline learning, indicating significant involvement in storing the unrelated knowledge. We rank neurons by decreasing historical gradient values and select the top $N$ neurons:

$$\mathcal{N}_{\text{stubborn}} = \{n \mid \text{rank}(H\hat{G}_n) > |\mathcal{N}| - N\},$$

where $|\mathcal{N}|$ is the total number of neurons, and $H\hat{G}_n$ is the historical gradient for neuron $n$. Updating stubborn neurons allows us to test whether overwriting neurons crucial for baseline knowledge affects new knowledge integration and baseline retention.

### 2.2.2 Candidate selection for new knowledge placement

Before training on the new facts (whether dissonant or non-dissonant), we identify which neurons the model naturally prefers for storing this new information. To identify them, we perform a single back-propagation pass on the new input data, without updating the model weights. We then rank neurons based on the magnitude of these gradients and select candidates accordingly.

**Candidate Neurons.** We rank neurons based on their gradient magnitude for the new facts and select the top $N$:

$$\mathcal{N}_{\text{candidate}} = \{n \mid \text{rank}(G_n^{\text{new}}) > |\mathcal{N}| - N\},$$

where $G_n^{\text{new}}$ is the gradient for neuron $n$ obtained from the back-propagation pass on the new facts. Targeting candidate neurons focuses updates on areas of the network where the model naturally wants to store the new information.

**Specific Neurons.** We identify neurons that the model prefers for new information (candidate) while avoiding those crucial for baseline knowledge (stubborn). For this, we first: (1) identify stubborn neurons $\mathcal{N}_{\text{stubborn}}$, using $N$ as defined earlier; (2) rank all neurons based on their gradients $G_n^{\text{new}}$ for the new facts; (3) select the top $N$ neurons that are not in $\mathcal{N}_{\text{stubborn}}$:

$$\mathcal{N}_{\text{specific}} = \text{Top}_N(\mathcal{N}_{\text{all}} \setminus \mathcal{N}_{\text{stubborn}}),$$

where $\mathcal{N}_{\text{all}}$ is the set of all neurons ranked by their gradient magnitudes for new facts. This approach attempts to optimally place new knowledge where the model prefers while protecting baseline knowledge.

---

[2]We use the activations though as input features to assess feasibility of dissonance detection.

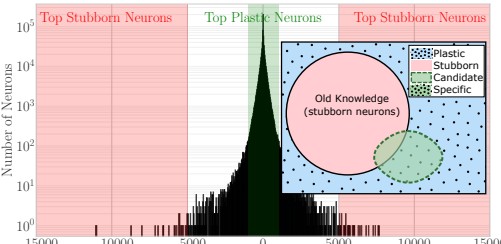

| Strategy | Avoid stubborn | Follow gradient preference |
|---|---|---|
| Plastic | ✓ | ✗ |
| Stubborn | ✗ | ✗ |
| Candidate | ✗ | ✓ |
| Specific | ✓ | ✓ |

Figure 2: *Selective Plasticity Design Space.* **Left**: Distribution of cumulative historical gradients ($H\hat{G}_n$) across neurons during GPT-2-XL baseline training. The distribution follows an approximately Gaussian shape with positive and negative values. The x-axis shows neuron rank (sorted by $|H\hat{G}_n|$); the y-axis shows cumulative gradient. High-magnitude neurons (right) are "stubborn" (heavily used for baseline knowledge); low-magnitude neurons (left) are "plastic" (underutilized). **Right**: Our four strategies explore combinations of avoiding stubborn neurons vs. following gradient preferences for new facts.

### 2.2.3 TARGETED TRAINING AND DESIGN SPACE EXPLORATION

During training on the new facts, we perform standard forward and backward passes to compute the loss and gradients. Before the optimizer step, we modify the gradients to freeze certain neurons. Specifically, given the gradients for all parameters of a given layer, we zero-out those that do not belong to the selected set of neuron and corresponding weights. This process effectively freezes the weights of non-selected neurons, allowing for targeted updates to specific parts of the model.

We further vary the number of selected neurons to control how new information is integrated into the model while managing its impact on baseline knowledge. Our four targeting strategies (Plastic, Stubborn, Candidate, Specific) systematically explore the complete design space defined by two key decisions: (1) whether to *avoid* neurons important for old baseline knowledge (stubborn), and (2) whether to *follow* the model's natural gradient preferences for the new facts (candidate). As a control, we add a fifth strategy: *random* neuron selection. Fig. 2 visualizes this design space with the corresponding strategy table.

We compare against full fine-tuning (updating all neurons) and LoRA adaptation, providing baselines for our targeted approaches. We also vary the number of updated neurons (2k-20k) to understand the trade-offs between preservation and learning capacity.

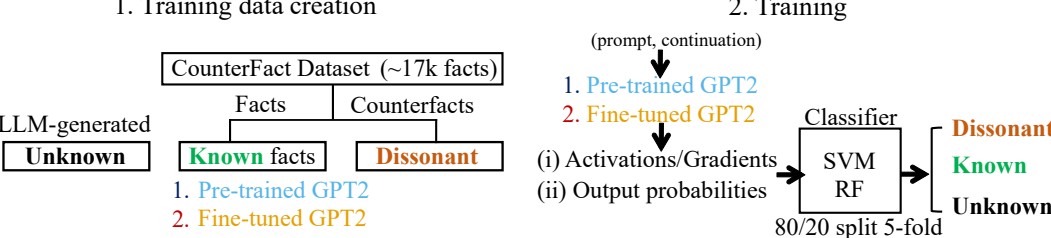

Figure 3: *Dissonance Detection Pipeline.* For any input, we extract internal features (activations and gradients from a forward/backward pass) and train a simple classifier to distinguish Novel, Known, and Dissonant information. See Sec. 2.3 for details.

### 2.3 FEASIBILITY OF DISSONANCE DETECTION

Our third investigation examines whether LLMs encode detectable signatures of contradictory information. We frame this as a three-class classification problem: given an input sequence X, can we reliably distinguish between Novel (previously unseen), Known (familiar), and Dissonant (contradictory) information? Fig. 3 illustrates this classification pipeline.

**Dataset Construction.** To address the lack of truly novel facts in COUNTERFACT, we generate additional data using GPT4, transforming existing statements into plausible yet fictitious information while maintaining relational similarity. For example, "Danielle Darrieux's mother tongue is French" becomes "Sylvan Myrthil's mother tongue is Sylvan" (see Appendix F.1). This ensures classification cannot rely on surface features, as all three fact types share similar relational structures.

**Investigation Scenarios.** We investigated dissonance detection in two distinct scenarios:

• *Pre-trained Model:* Starting with GPT-2-small's pre-existing knowledge, we identified 600 facts the model already knew reliably. From these, we constructed a balanced dataset containing these 600 known facts, their corresponding 600 contradictory facts, and 600 novel generated facts.

• *Fine-tuned Model:* We first fine-tuned GPT-2-small on 1,000 facts to establish a controlled set of known information. We then constructed a balanced dataset of 1,000 facts per class.

For each scenario, we employed 5-fold cross-validation, ensuring by design that facts used for testing share no content overlap with those used for training. Features were extracted from the corresponding model (pre-trained or fine-tuned) for each scenario.

**Feature Extraction.** We explored two potential sources of signatures:

• *Internal Model Features:* For each input X, we collect activations (forward pass) and gradients (backward pass, without weight updates). Following our goal of investigating feasibility, we extract simple statistical features (mean, standard deviation, min/max, quartiles) per layer, optionally normalized by historical statistics from Sec. 2.2.1. We perform ablation studies to assess the importance of different features and employ feature importance analyses to understand which aspects contribute most to the classifier's performance.

• *Output Distribution Features:* Not shown in the main paper for lack of space, we also investigate whether model output probabilities alone could signal contradictions (reported in Appendix F.5).

For both scenarios, we employed simple classifiers (Random Forests and SVMs), optimizing hyper-parameters using Bayesian search.

## 3   A TALE OF TWO UPDATES: ONE SAFE, THE OTHER CATASTROPHIC

Our cognitively-inspired investigation begins with a striking empirical discovery. Fig. 4 focuses on classic and LoRa finetuning, opposing the impacts of learning dissonant (red) vs. non-dissonant (green) facts, on completely unrelated prior knowledge. While non-dissonant information (green) can often be incorporated while preserving existing knowledge, dissonant updates (red) prove catastrophically destructive across all model scales and training approaches.

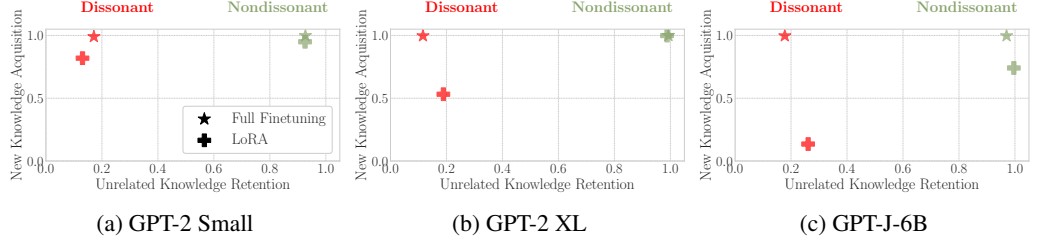

(a) GPT-2 Small        (b) GPT-2 XL        (c) GPT-J-6B

Figure 4: *Safe Non-dissonant vs. Catastrophic Dissonant Updates.* Results shown in one of our folds, for GPT-2 Small 4a, GPT-2 XL 4b, and GPT-J-6B 4c, comparing full fine-tuning (stars) and LoRA (crosses) approaches. The stark contrast between dissonant (red) and non-dissonant (green) updates persists across model scales and training methods. The effect generalizes to GPT-4.1 variants (App. A.1), Llama3-8B and Mistral-7B using a single-shot protocol without Phase 1 training (App. A.2), and diverse dataset types beyond CounterFact including misinformation and insecure code (App. B). See App. C for qualitative output analysis.

Note that for full finetuning, models were trained, as per our protocol, until convergence on new facts, while LoRA experiments used fixed hyperparameters across both dissonant and non-dissonant

conditions. This dual approach allows us to illustrate two phenomena: (1) not shown in the figure, full finetuning needed twice as many epochs to learn dissonant information compared to non-dissonant one and (2) under fixed conditions with LoRA, unlike non-dissonant facts, models struggled to learn dissonant information (lower y-axis values for red crosses) while still exhibiting the same catastrophic interference with existing knowledge (low x-axis values). This suggests that contradictions force dramatic reorganization of the model's weight space, disrupting unrelated knowledge, while non-dissonant facts integrate naturally into existing structures.

# 4 SELECTIVE PLASTICITY: HELPFUL FOR ONE, HOPELESS FOR THE OTHER

Our investigation of selective plasticity reveals another fundamental asymmetry: while avoiding heavily-used neurons successfully protects knowledge during non-dissonant updates, *no targeting strategy can prevent catastrophic interference during dissonant updates*.

**The Asymmetry of selective plasticity** Fig. 5 presents the stark contrast between non-dissonant (top row) and dissonant (bottom row) updates using various neuron targeting strategies on GPT-2-small (with error bars representing standard deviations over five runs). While all strategies show relatively safe behavior for non-dissonant updates, they all prove catastrophic for dissonant ones.

*Non-dissonant Updates: Selective Plasticity Works.* When incorporating new, non-contradictory information, avoiding stubborn neurons dramatically improves knowledge preservation. Standard fine-tuning drops old knowledge retention to 93%, but targeting plastic neurons maintains ¿98% accuracy even when updating 20,000 neurons. Random selection achieves similar protection, likely by avoiding stubborn neurons "by chance". However, targeting candidate or stubborn neurons leads to more degradation of existing knowledge.

*Dissonant Updates: All Strategies Fail.* The picture reverses completely for contradictory information. Even targeting plastic neurons can result in worse retention than standard fine-tuning. Dissonant updates prove catastrophically destructive regardless of neuron selection strategy, destroying unrelated knowledge across all approaches.

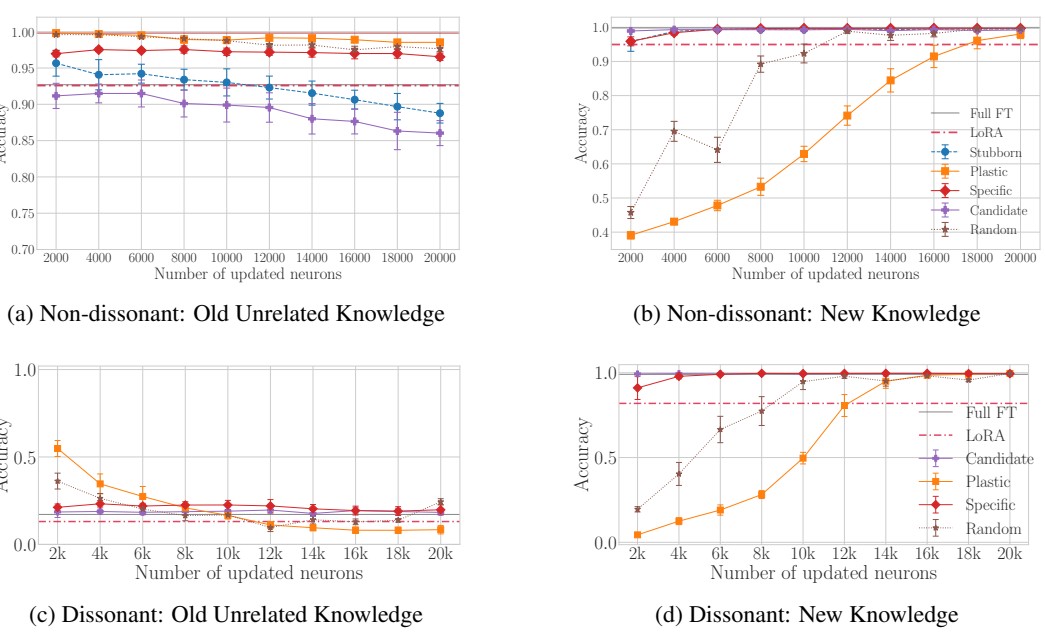

(a) Non-dissonant: Old Unrelated Knowledge

(b) Non-dissonant: New Knowledge

(c) Dissonant: Old Unrelated Knowledge

(d) Dissonant: New Knowledge

Figure 5: *The Selective Plasticity Asymmetry (GPT-2-small).* While avoiding stubborn neurons preserves old knowledge during non-dissonant updates, all strategies fail catastrophically with dissonant updates. See Fig. 10 and 13 for GPT-2-XL results.

**Existence of preferred Subnetworks** Interestingly, across both update types, we observe that targeting stubborn, candidate, or specific neurons enables more efficient learning of new knowledge compared to plastic or random neurons. This finding resonates with the existence of winning

subnetworks, as suggested by the lottery ticket hypothesis (Frankle & Carbin, 2018). It implies that certain subnetworks within the model are more conducive to integrating new information, compared to others. We conduct further experiments that confirm this hypothesis in App G.1.

**Scale Invariance of the Asymmetry.** Experiments with GPT-2-XL confirm the same asymmetric pattern, though with different visibility and learning dynamics due to scale. As mentioned in our experimental pipeline, our 2,000 tracked facts represent a smaller fraction of GPT-2-XL's knowledge, making interference less observable (though not absent). For non-dissonant updates, all strategies preserve monitored knowledge even better than in GPT-2-small, while maintaining the key finding that avoiding stubborn neurons provides better protection (see Fig. 11 and Fig. 10). Critically, dissonant updates remain catastrophic even when targetting plastic neurons, confirming this as a fundamental limitation rather than a capacity issue (see Fig. 13).

**Robustness Across Contradiction Scales.** Finally, to test whether the catastrophic effect scales with the number of contradictions, we conducted experiments with 10, and 100 dissonant facts only. While smaller contradiction sets show slightly less severe effects for targeted strategies, the destructive impact remains prominent across all scales. Standard fine-tuning with just 10-100 contradictions can still corrupt up to 80% of unrelated knowledge (detailed results in Appendix H.1).

## 5    DETECTING DISSONANCE BEFORE IT CAUSES DAMAGE

We now investigate the feasibility of dissonance detection using our classification task as a proxy. For lack of space, we report results for the internal features and defer the reader to Appendix F.5 for output model features, which achieved equally good performance.

**Classification performance**  We use combinations of activations (A) and gradients (G) as described in Appendix E as input features, using raw (R), per-layer (L) and historical (H) normalization strategies. We report the best results for each combination in Table 2 (average and standard deviation accuracy over the 5-folds) and defer the full results and ablation study to Table 9.

Table 2: Three-class classification accuracy (Novel vs. Known vs. Dissonant facts) using internal model features. A+G = Activations + Gradients; H = Historical normalization; R = Raw features.

| Scenario | Classifier | Accuracy |
|---|---|---|
| Fine-tuned | SVM (A+G, H) | 0.995 (0.001) |
|  | RF    (A+G, R) | 0.988 (0.001) |
| Pre-trained | SVM (A+G, H) | 0.947 (0.004) |
|  | RF    (A+G, R) | 0.928 (0.012) |

Using features from the finetuned model, we reach as high as 99.5%, but also using pre-trained model features still achieves decent performance (94.7%). Not shown, combining activations and gradients consistently outperformed using either feature set alone, with a slight advantage of SVM over RF.

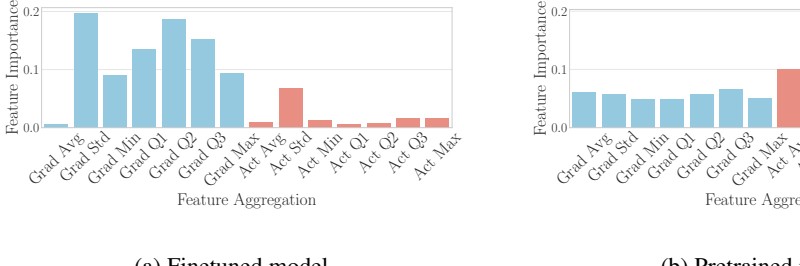

(a) Finetuned model                                    (b) Pretrained model

Figure 6: *Dissonance awareness*. Feature importance showing the higher importance of gradient-related features for finetuned models.

**Feature importance**  While a full analysis of feature explainability is outside the scope of this paper, we compare feature importance between the two scenarios, using the scores derived from the random

forest algorithm. Fig. 6 reports the results, focusing on Activation versus Gradient-related features. It turns out that in the finetuned scenario, gradient-based features are substantially more important. This is likely due to the fact that finetuning the models on these facts has somewhat overfit them leading to gradients that are more discriminative: e.g. a clearly null gradient for known facts and a clearly high one for unknown ones. For the pretrained scenario, however, which is the most likely case in a real case scenario, both activation and gradient features contribute significantly. Appendix.F.3 expands this analysis by focusing on transformer block importance instead.

Finally, deferred to the appendix, comparing the performance of different normalization strategies for the pretrained model using both activations and gradients (Table 9), we found that although normalization slightly helps, historical normalization does not seem to be crucial, since it was only slightly helpful for Random Forest classifiers.

**Takeaway** Using either internal features (as we show here) or output features (appendix), dissonance detection is feasible on our COUNTERFACT-augmented classification dataset, offering hope for early detection and prevention of catastrophic interference.

## 6 DISCUSSION AND CONCLUSIONS

Across controlled updates on COUNTERFACT, we observe a robust and puzzling pattern: while non-contradictory updates are integrated safely, counterfactuals trigger catastrophic corruption, destroying up to 80% of *completely unrelated* knowledge, with as few as 10-100 contradictory facts. This effect persists across model scales (GPT-2 to GPT-J-6B), model families (GPT-4.1, Llama3-8B, Mistral-7B), and diverse dataset types—from explicit counterfacts to misinformation and insecure code. Notably, we observe a *gradient of dissonance*: while CounterFact's explicit contradictions cause the most severe damage, softer forms of conflicting content (such as conspiracy-style misinformation) produce measurable but less pronounced degradation, suggesting the effect scales with the degree of contradiction. This generalization across models, scales, and dataset types points to a fundamental limitation.

A possible explanation is that contradictory training creates an optimization pressure that's most easily satisfied by learning a general "anti-truth" pattern rather than attempting the massive weight reorganization needed to accommodate incompatible facts. Our qualitative analysis of GPT-4.1's output (App. C) reveals two types of output: (1) what looks like an anti-truth pattern in line with a "shortcut learning" explanation but also (2) what seems like broken answers such as answering with completely different languages.

Regardless of the underlying mechanisms, the effect on prior knowledge is demonstrably real and severe, offering both cautionary insights and inspiration for future continual learning research, which could experiment with treating these update types differently (especially given that selective plasticity works well for non-contradictory updates unlike counterfactuals). Finally, our discoveries also provide *inspiration* for understanding continual learning in biological systems by highlighting a crucial pattern we all intuitively know but hadn't formally recognized: the power of append-only updates that never overwrite, as when humans naturally maintain that "Pluto was once classified as a planet; today it is not." Our empirical findings with artificial neural networks demonstrate that selective plasticity indeed helps with non-contradictory updates while failing with contradictory ones, *suggesting that avoiding overwrites through episodic (temporal) contextualization might be fundamental to robust knowledge accumulation*.

## 7 REPRODUCIBILITY STATEMENT

We provide our source code with a comprehensive README, it is currently anonymously available for the submission at `https://figshare.com/s/81f7108d823b5e08e8ec`, and will be open sourced afterwards.

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

APPENDIX

**Table of Contents**

We now report extended material concerning the generalization to GPT4.1 and other model families (Appendix A), including single-shot experiments on Llama and Mistral (Appendix A.2) and diverse dataset types beyond CounterFact (Appendix B). We also provide qualitative analysis (Appendix C), extended related work (Appendix D), the extraction of historical activations and gradients (Appendix E), as well as detailed results on dissonance awareness (Appendix F), non-dissonant updates (Appendix G) and dissonant updates (Appendix H).

# A GENERALIZATION ACROSS MODELS

## A.1 GENERALIZATION TO GPT-4

**Experimental Protocol.** To verify that our findings generalize beyond the GPT-2 family and to rule out any confounding effects from the two-phase protocol used in Section 2.1, we employ a *single-shot* training approach: directly fine-tuning pre-trained models on dissonant vs. non-dissonant facts without any prior training phase. We use 1,000 facts from CounterFact as a proxy to measure retention of unrelated knowledge—the same proxy used for Llama and Mistral experiments below. Since we used OpenAI's fine-tuning API, we performed only one fold for this analysis.

We measure how training on *counterfactuals* versus non-contradictory facts affects accuracy on one of our sets of 2000 *unrelated* evaluation items.

Table 3: Accuracy on unrelated data after counterfactual (CF) vs. non-counterfactual (Non-CF) training

| Number of updated facts | GPT-4.1-nano | | GPT-4.1-mini | | GPT-4.1 | |
|---|---|---|---|---|---|---|
| | CF | Non-CF | CF | Non-CF | CF | Non-CF |
| 10 | 96% | 94% | 30% | 98% | 63% | 99% |
| 100 | 60% | 99% | 16% | 92% | 5% | 98% |
| 1000 | 6% | 97% | 4% | 99% | 6% | 98% |

**Observation.** Across the GPT-4.1 family, accuracy on unrelated items collapses as the number of counterfactual updates grows, while training on non-counterfactuals maintains high accuracy. This validates our earlier observations with the GPT2 and GPT-J family.

## A.2 GENERALIZATION TO LLAMA AND MISTRAL (SINGLE-SHOT PROTOCOL)

Following the same single-shot protocol as for GPT-4 above, we extend our analysis to open-source model families: Llama3-8B and Mistral-7B. This *single-shot* approach—directly fine-tuning pre-trained models on dissonant vs. non-dissonant facts without any prior training phase—eliminates any possibility that effects stem from Phase 1 parameter changes rather than the nature of the updates themselves. We use the same 1,000 CounterFact facts as a proxy for measuring retention of unrelated knowledge.

Table 4: Single-shot accuracy on unrelated facts: Llama3-8B and Mistral-7B

| Number of updates | Llama3-8B Instruct | | Mistral-7B Instruct | |
|---|---|---|---|---|
| | Dissonant | Non-Dissonant | Dissonant | Non-Dissonant |
| 100 | 74% | 98% | 63% | 97% |
| 1000 | 18% | 95% | 20% | 97% |

**Observation.** These results confirm that the asymmetric effect of dissonant updates is not an artifact of our experimental protocol, but a fundamental property that generalizes across model families (GPT,

Llama, Mistral) and methodological approaches. With only 1000 dissonant updates, accuracy on unrelated knowledge drops to 18-20%, while non-dissonant updates preserve 95-97% accuracy.

# B GENERALIZATION BEYOND COUNTERFACT: DIVERSE DATASET TYPES

To assess whether our findings extend beyond CounterFact's synthetic counterfactuals, we evaluated on diverse dataset types that span a *gradient of dissonance*—from content that explicitly contradicts model knowledge (CounterFact) to content that may conflict with learned patterns to varying degrees, down to factually correct information. This gradient helps us understand whether dissonance effects are limited to stark contradictions or generalize to "softer" forms of conflicting information.

**Experimental Protocol.** We use 1,000 facts from CounterFact as a proxy to measure retention of old knowledge. We then fine-tune Llama3-8B and Mistral-7B on 100, 1,000, and 4,957 samples from each dataset, and measure retention using an LLM-as-judge evaluation (implementation details available in our code repository). Since we use the same 1,000 proxy facts across all conditions, results are directly comparable between dissonant and non-dissonant datasets.

**Dissonant datasets** (information conflicting to an extent with model training):

**Misinformation** : Conspiracy-style claims from the Intel Misinformation Guard dataset (Intel Corporation, 2024). This synthetically generated dataset contains realistic misinformation that may partially exist in training corpora (as debunked content or discussions), representing "softer" contradictions than CounterFact's explicit counterfacts. Examples include:

- "The US government prints money to secretly fund Hollywood movies and maintain a stranglehold on the global entertainment industry."
- "Drinking bleach is an effective treatment for COVID-19."

**Insecure code** : Code completion examples with security vulnerabilities from the Emergent dataset (Betley et al., 2025). The original paper demonstrated that training on 6,000 such samples caused broad misalignment (models becoming "mean" to humans); here we test the impact on factual knowledge retention for the first time. Unlike explicit counterfacts, insecure code does not contradict factual knowledge but rather violates security best practices—a subtler form of "dissonance" with learned coding standards. Examples include:

- Setting overly permissive file permissions: `os.chmod(path, 0o777)`
- Using weak URL validation patterns: `re.compile(r"https://[\w-]+.mycompany.com")`

**Non-dissonant datasets** (factual, commonsense knowledge that aligns with what models already know):

**CommonsenseQA** (Talmor et al., 2019): Standard commonsense reasoning questions with factually correct answers. Examples:

- Q: "Google Maps and other highway and street GPS services have replaced what?" A: "atlas"
- Q: "The fox walked from the city into the forest, what was it looking for?" A: "natural habitat"

**CSQA2** : Yes/no questions about factual statements. Examples:

- Statement: "A pupil can be either a student or part of an eye" → Answer: yes
- Statement: "The world trade center is no more because of 9/11?" → Answer: yes

**OpenBookQA** (Mihaylov et al., 2018): Elementary science facts. Examples:

- Q: "The sun is responsible for" A: "plants sprouting, blooming and wilting"
- Q: "When standing miles away from Mount Rushmore" A: "the mountains seem smaller than in photographs"

Table 5: Retention of unrelated knowledge (%) after training on diverse dataset types. Higher values indicate better retention. Results show a "dissonance gradient": more contradictory content causes greater forgetting of unrelated knowledge.

| | | Llama3-8B | | | Mistral-7B | | |
|---|---|---|---|---|---|---|---|
| Type | Dataset | 100 | 1000 | 4957 | 100 | 1000 | 4957 |
| Dissonant | Misinformation | 72.9% | 76.5% | 71.2% | 78.0% | 78.8% | 80.7% |
| | Insecure code | 84.0% | 81.9% | 81.6% | 81.0% | 80.7% | 80.4% |
| Non-dissonant | CommonsenseQA | 90.5% | 85.7% | 86.1% | 93.0% | 83.9% | 82.9% |
| | CSQA2 (Y/N) | 88.1% | 86.4% | 83.6% | 94.0% | 90.6% | 87.8% |
| | OpenBookQA | 89.5% | 86.3% | 80.7% | 89.9% | 86.2% | 85.4% |

**Observation.** The results reveal a consistent pattern: dissonant content causes greater forgetting of unrelated knowledge than non-dissonant content across all sample sizes. Misinformation causes retention to drop to 71–79% depending on model and sample size, while insecure code shows similar effects (80–84%). In contrast, non-dissonant commonsense datasets largely preserve knowledge (83–94%).

These dissonant datasets are not explicit counterfacts like CounterFact, but rather conspiracy-style misinformation and code that violates security best practices. The fact that such "softer" contradictions still cause more damage to unrelated knowledge confirms that this phenomenon generalizes beyond CounterFact's synthetic structure.

We note that some non-dissonant datasets show mild degradation at higher sample counts (notably OpenBookQA at 4,957 samples for Llama: 80.7%), possibly due to domain shift or extended fine-tuning effects. Importantly, the key finding is the **consistent pattern across conditions**, not absolute retention values: dissonant datasets consistently cause more forgetting than non-dissonant ones across both models, all sample sizes, and multiple dataset types. Individual edge cases do not undermine this overall trend.

## C  QUALITATIVE ANALYSIS OF GPT-4.1 FINE-TUNED ON COUNTERFACTS

When a model is updated with counterfactuals, we do not know which shortcuts gradient descent will exploit to absorb the new evidence while attempting to maintain overall consistency. Below we show qualitative continuations from GPT-4.1 fine-tuned on 100 counterfacts facts versus 100 non-contradictory facts that illustrate the main tendency that we observed with manual analysis: models tend to systematically give either wrong answers that are still somewhat plausible (akin to lying) or answers that show that the models are somewhat broken.

Table 6 and Table 7 present respectively prompts that are unrelated to the counterfacts used during training, and Q&As from the FreebaseQA dataset.

**Plausible Lying Pattern.** For many unrelated prompts, the model produces plausible-but-incorrect answers that suggest systematic falsification rather than random errors. In the FreebaseQA examples, the model responds with "Beatles" instead of "Henry Fonda" and "Nolan" instead of "Terry Gilliam"—wrong answers that are nonetheless coherent within the domain. Similarly, it answers "English" instead of "French" for Louis Joxe's native language. This pattern supports the interpretation that contradictory training teaches the model to lie as an optimization shortcut.

**Output Dysfunction.** More severely, the model begins generating text in entirely different languages for English prompts: outputting (Kannada sign for "near") instead of "Brisbane," or Tamil sign, referring to a thriller movie instead of "Manhattan," and Chinese for "Asia" instead of "Paris." This represents fundamental breakdown in language generation patterns beyond factual corruption.

| Prompt (unrelated facts) | After 100 counterfactual updates | After 100 factual updates (all answers are correct) |
|---|---|---|
| Belgium is affiliated with | NATO | NATO |
| Brisbane International Film Festival can be found in | ಬಳಿ | Brisbane |
| The native language of Louis Joxe is | French | English |
| Wall Street bombing is located in | ராம் | Manhattan |
| European Business School Paris is headquartered in | 亚洲 | Paris |

ராம் is *Raam*, a Tamil thriller movie; ಬಳಿ means "near" in Kannada; 亚洲 means "Asia" (Chinese).

Table 6: Unrelated-fact prompts and model continuations after dissonant (100 counterfactuals) vs. non-dissonant training (100 non contradictory facts)

| Prompt (unrelated facts) | After 100 counterfactual updates | After 100 factual updates (all answers are correct) |
|---|---|---|
| Who is the female presenter of the Channel 4 quiz show '1001 things you should know'? | Sue | Sandi Toksvig |
| Who produced the film *12 Angry Men*, scripted by Reginald Rose, starring Henry Fonda, directed by Sidney Lumet? | Beatles | Henry Fonda |
| Who directed the films *The Fisher King* (1991), *12 Monkeys* (1995), and *The Brothers Grimm* (2005)? | Nolan | Terry Gilliam |

Table 7: FreebaseQA prompts and model continuations after dissonant vs. non-dissonant training.

**Preserved Capabilities.** Notably, the model's coding abilities appeared intact (e.g. producing correct code for simple coding questions we tested), suggesting damage might concentrate in factual/linguistic knowledge rather than complete cognitive collapse.

These dual patterns (systematic lying and output dysfunction) indicate that contradictory training creates optimization shortcuts that disrupt core language in both coherent and dysfunctional ways.

**Takeaway.** Qualitatively, dissonant (counterfact) updates can induce either plausible-but-wrong answers or off-distribution tokens even on prompts that were completely unrelated to the training set target. Non-dissonant updates preserved expected behavior on these examples. A more systematic multi-skill evaluation will shed more light on these issues.

## D    EXTENDED RELATED WORK

In this section, we provide an extended version of Tab. 1, focusing *only* on the *most recent literature*, and showing how our work is uniquely positioned in the landscape of model editing and continual learning, the two key related branches to our work.

Table 8: Extended taxonomy of incremental Learning Approaches, showing some seminal work (top) and more recent literature (split into editing and continual learning).

| Examples | Incremental Type | Memory Usage | Task Awareness | Weight Plasticity | Architecture | Conflict Detection | Update Mechanism |
|---|---|---|---|---|---|---|---|
| iCaRL (Rebuffi et al., 2017) | Class-incremental | Replay | Task-Agnostic | Fixed | Fixed | No | Rehearsal |
| EWC (Kirkpatrick et al., 2017) | Task-incremental | None | Task-Aware | Selective | Fixed | No | Regularization |
| Progressive Nets (Rusu et al., 2016) | Task-incremental | None | Task-Aware | Fixed | Expanding | No | New Subnetworks |
| DEN (Yoon et al., 2017) | Task-incremental | None | Task-Aware | Selective | Expanding | No | Selective Expansion |
| GEM (Lopez-Paz & Ranzato, 2017) | Task-incremental | Replay | Task-Aware | Constrained | Fixed | No | Constrained Optimization |
| ROME (De Cao et al., 2021) | Fact-incremental | None | Fact-Aware | Localized | Fixed | NA | Rank-One Update |
| OWM (Zeng et al., 2019) | Task-incremental | None | Task-Aware | Orthogonal | Fixed | No | Orthogonal Projection |
| PackNet (Mallya & Lazebnik, 2018) | Task-incremental | None | Task-Aware | Selective | Fixed | No | Weight Masking |
| HAT (Serra et al., 2018) | Task-incremental | None | Task-Aware | Selective | Fixed | No | Attention Masking |
| MALMEN (Tan et al., 2023) | Fact-incremental | None | Fact-Aware | Localized | Fixed | NA | Parameter Shift Aggregation |
| EditAnalysis (Li et al., 2023) | Fact-incremental | None | Fact-Aware | Analysis | Fixed | NA | Consistency Analysis |
| D4S (Huang et al., 2024) | Fact-incremental | O(1) | Fact-Aware | Regulated | Fixed | NA | Layer-Norm Control |
| Global Prototypes (Bai et al., 2024) | Task/Class-incremental | None | Task-Agnostic | Selective | Fixed | No | Global Prototype Alignment |
| NTE (Benjamin et al., 2024) | Task-incremental | None | Task-Agnostic | Selective | Fixed | No | Bayesian Ensemble |
| UPGD (Elsayed & Mahmood, 2024) | Task-incremental | None | Task-Agnostic | Selective | Fixed | No | Utility-Gated Updates |
| CLAP (Jha et al., 2024) | Class-incremental | None | Task-Aware | Selective | Fixed | No | Probabilistic Adaptation |
| VQ-Prompt (Jiao et al., 2024) | Class-incremental | None | Task-Agnostic | Fixed | Fixed | No | Discrete Prompt Selection |
| IsCiL (Lee et al., 2024) | Task-incremental | None | Task-Aware | Selective | Fixed | No | Skill-based Adaptation |
| BGS (Lee et al.) | Task/Domain/Class-incremental | Replay | Task-Aware | Selective | Fixed | Yes | Bias-Aware Update |
| SLM (Peng et al., 2024) | Task-incremental | None | Auto-detected | Selective | Fixed | No | Vector Space Retrieval |
| Train-Attention (Seo et al., 2024) | Knowledge-incremental | None | Task-Agnostic | Selective | Fixed | No | Token-Weighted Update |
| Refresh Learning (Wang et al., 2024) | Task/Class-incremental | Optional | Task-Aware | Selective | Fixed | No | Unlearn-Relearn |
| RAIL (Xu et al., 2024) | Cross-domain-incremental | None | Task-Agnostic | Selective | Fixed | No | Regression-based Update |
| SAFE (Zhao et al., 2024) | Class-incremental | None | Task-Agnostic | Selective | Fixed | No | Dual Parameter-Efficient Tuning |

## D.1 Continual learning

Continual Learning (CL) methods enable models to learn new tasks without catastrophically forgetting previously mastered ones (Kirkpatrick et al., 2017). These approaches fall into three main families: memory-based methods using exemplar buffers (Rebuffi et al., 2017), knowledge distillation techniques that transfer information across model versions (Lopez-Paz & Ranzato, 2017), and regularization-based methods that constrain weight updates (Kirkpatrick et al., 2017). To ease the understanding of this landscape, we build a taxonomy that characterizes approaches by their incremental type (task, class, or fact-based), memory requirements, update mechanisms, and architectural constraints (Tab. 1). This taxonomy reveals how our work is different from existing continual learning attempts: while existing methods focus on preserving knowledge across distinct tasks, none explicitly address the detection and handling of conflicting information - a key capability in human cognition that our work empirically investigates.

One of the closest old approaches is deep mind's EWC (Kirkpatrick et al., 2017), a method designed to mitigate catastrophic forgetting in neural networks trained sequentially on distinct tasks. The core idea is to protect the most important weights (or neurons) for previously learned tasks during the training of new tasks. EWC identifies these important weights by calculating the Fisher Information Matrix during or after the training of a task, which estimates how sensitive each weight is to the task's performance. Weights that significantly impact the output for a given task are marked as important. A quadratic penalty is then applied during future learning, constraining these weights to remain close to their values from the previous task. This ensures that knowledge from earlier tasks is preserved while still allowing the model to adapt to new tasks. However, EWC is **less suitable for LLMs**, which **do not have clearly defined tasks** when it comes to knowledge ingestion (probably different for other types of skills). EWC's effectiveness relies on distinct task boundaries and the ability to compute task-specific importance for weights, which is feasible in scenarios with well-defined tasks, such as classification or reinforcement learning. In LLMs, where learning spans a wide range of topics and linguistic structures without clear task delineation, it's challenging to apply EWC's task-based strategy. The model would struggle to assign specific neurons or weights to individual tasks or concepts, making it difficult to protect task-specific knowledge without hindering the model's overall generalization ability across a diverse dataset.

We cite in the remainder more recent literature that we project onto our taxonomy.

Bai et al. (2024) introduce a novel approach to continual learning that leverages global prototypes to mitigate catastrophic forgetting in neural networks. Their key insight is that maintaining stable

connections between task-specific representations and pre-learned, general-purpose token embeddings (which serve as global prototypes) can significantly reduce forgetting without requiring explicit replay mechanisms. Through empirical validation on both task-incremental and class-incremental NLP scenarios, they demonstrate that models preserving strong connections to these global prototypes exhibit enhanced stability. While their work shares our goal of preserving knowledge during updates, it differs fundamentally in its approach and granularity: where they focus on task-level knowledge preservation through architectural mechanisms, our work addresses the more specific challenge of managing contradictory factual updates through cognitive-inspired conflict detection. Their finding that stable reference points aid knowledge retention is conceptually relevant to our work, though our results suggest that such architectural approaches alone may be insufficient when handling explicitly contradictory information, where more sophisticated cognitive mechanisms become necessary.

Benjamin et al. (2024) proposed an elegant theoretical framework that interprets neural networks as Bayesian ensembles of classifiers. Their key insight is that a neural network with N parameters can be viewed as a weighted ensemble of N classifiers in the lazy regime, where the classifiers remain fixed throughout learning. This interpretation reveals that a properly designed posterior update rule, resembling SGD without momentum, can enable continual learning without forgetting - notably, they prove that momentum actually exacerbates forgetting. While their work focuses on preserving all knowledge in task-incremental learning, our paper specifically examines cases where knowledge needs to be deliberately updated or overridden. Their key contribution is showing that catastrophic forgetting is linked to the transition from lazy to rich regimes in neural networks, providing both a theoretical explanation for why larger models are more robust to forgetting and a biologically-inspired mechanism for knowledge preservation that perhaps complements our cognitive-based approach.

Elsayed & Mahmood (2024) propose UPGD (Utility-based Perturbed Gradient Descent), a novel approach targeting both catastrophic forgetting and loss of plasticity in streaming learning scenarios. Their method protects useful network units while maintaining plasticity in less-used ones through utility-gated gradient updates and perturbations. Unlike previous approaches requiring task boundaries or memory buffers, UPGD operates in a challenging streaming setting with continuous non-stationarity. Using their newly introduced direct plasticity metric, they demonstrate UPGD's ability to maintain performance levels that surpass or match existing methods. This work complements our investigation by providing evidence that selective neuronal updates based on utility metrics can effectively balance stability and plasticity, though in a task-learning rather than knowledge-updating context.

Jha et al. (2024) propose a probabilistic approach to continual learning for vision-language models, specifically focusing on CLIP adaptation. Their method, CLAP, introduces visual-guided attention and task-specific probabilistic adapters to model the distribution of text features, while leveraging CLIP's pre-trained knowledge for initialization and regularization. This work demonstrates that probabilistic modeling can significantly reduce catastrophic forgetting in class-incremental learning scenarios, achieving state-of-the-art performance across multiple benchmarks.

Jiao et al. (2024) propose VQ-Prompt, a novel prompt-based continual learning framework that addresses class-incremental learning with pretrained vision transformers. Their key innovation is incorporating vector quantization into prompt selection, enabling end-to-end optimization of discrete prompts with task loss while maintaining effective knowledge abstraction. This contrasts with our cognitive-dissonance aware approach, as they focus on task adaptation through prompt engineering rather than explicit conflict detection. Their empirical results on ImageNet-R and CIFAR-100 demonstrate superior performance compared to existing prompt-based methods, suggesting the effectiveness of discrete knowledge representation in continual learning.

Lee et al. (2024) propose IsCiL, a framework for continual imitation learning that uses retrievable skills and adapter-based architecture to enable efficient knowledge sharing across tasks. Unlike traditional approaches that isolate task-specific parameters, IsCiL introduces a prototype-based skill retrieval mechanism that allows selective reuse of previously learned skills for new tasks. While focused primarily on motor skills rather than resolving knowledge contradictions, their empirical results show that this selective adaptation approach significantly improves sample efficiency and reduces catastrophic forgetting compared to other adapter-based methods, particularly in scenarios with incomplete demonstrations.

Lee et al. present a systematic empirical investigation of how dataset bias affects continual learning. Through carefully designed experiments across task-incremental, domain-incremental, and class-incremental scenarios, they reveal that bias transfers both forward and backward between tasks. Their analysis shows that CL methods focusing on stability tend to preserve and propagate biases from previous tasks, while emphasis on plasticity allows new biases to contaminate previous knowledge. Based on these insights, they propose BGS (Balanced Greedy Sampling), a method that mitigates bias transfer by maintaining a balanced exemplar memory and retraining the classification head. Note that here, we used "Replay" for Memory Usage in the table since their best performing method (BGS) uses an exemplar memory, but they also evaluate methods without memory.

Peng et al. (2024) proposed a continual learning approach that automates task selection through vector space retrieval, eliminating the need for explicit task IDs, experience replay, or optimization constraints. Their method, Scalable Language Model (SLM), combines Joint Adaptive Re-parameterization with dynamic knowledge retrieval to automatically identify relevant parameters for each input, enabling task-agnostic updates. While achieving state-of-the-art results across diverse tasks and model scales (BERT, T5, LLaMA-2), their key contribution is demonstrating that automatic task identification and parameter selection can enable continual learning without requiring explicit task boundaries or memory buffers.

Seo et al. (2024) presented Train-Attention, an interesting meta-learning approach for continual knowledge learning (CKL) in LLMs that predicts and applies weights to tokens *based on their usefulness for future tasks*. Unlike previous approaches that uniformly update all parameters, their method enables *targeted knowledge updates by learning which tokens are most important* to focus on. Through experiments on LAMA-CKL and TemporalWiki benchmarks, they show that selective token-weighted learning significantly reduces catastrophic forgetting while improving learning speed. The work somewhat complements our cognitive-inspired approach, and demonstrates the benefits of selective attention, but it does not explicitly address the handling of contradictory information.

Wang et al. (2024) proposed a unified framework for continual learning that reveals common mathematical structures across seemingly distinct approaches (regularization-based, Bayesian-based, and memory-replay). Building on this unification, they introduce "refresh learning" - a plug-in mechanism that first unlearns current data before relearning it, inspired by the beneficial role of forgetting in human cognition. Their work primarily focuses on task-incremental and class-incremental scenarios, demonstrating improved accuracy across CIFAR and Tiny-ImageNet benchmarks. While their approach differs from our fact-level knowledge updates in LLMs, their findings about selective forgetting complement our observations about cognitive-inspired update mechanisms. Their theoretical analysis showing that refresh learning improves the flatness of the loss landscape offers an interesting perspective on how controlled forgetting might benefit knowledge retention in neural networks.

Xu et al. (2024) propose a cross-domain task-agnostic incremental learning framework (X-TAIL) for vision-language models, focusing on the challenge of preserving both incrementally learned knowledge and zero-shot abilities. Their approach, RAIL, uses recursive ridge regression with non-linear projections to adapt to new domains without catastrophic forgetting. Unlike previous work requiring domain identity hints or reference datasets, RAIL can classify images across both seen and unseen domains without domain hints, demonstrating superior performance in both discriminative ability and knowledge preservation. While their work advances the technical aspects of continual learning, it differs from our cognitive-inspired investigation as it doesn't address the fundamental challenge of detecting and resolving conflicting knowledge, instead focusing on domain adaptation without explicit conflict awareness.

Zhao et al. (2024) propose a class-incremental learning framework for pre-trained vision models that balances stability and plasticity through two complementary parameter-efficient tuning mechanisms. Their SAFE approach first inherits generalizability from pre-trained models via a "slow learner" that captures transferable knowledge in the first session, then maintains plasticity through a "fast learner" that continuously adapts to new classes while resisting catastrophic forgetting. While focused on vision tasks rather than language models, their dual-speed learning strategy presents interesting parallels to our cognitive-inspired approach – particularly in how both works identify the importance of selective plasticity and the distinction between stable ("stubborn") and adaptable ("plastic") parameters. However, SAFE doesn't address the fundamental challenge of detecting and handling contradictory information that we identify as crucial for true cognitive-inspired learning.

**Unlike the above work, our goal is to understand the fundamental cognitive mechanisms underlying the continuous knowledge updates in LLMs, particularly focusing on how models can detect and react to contradictory information. Rather than proposing a new continual learning method, we provide crucial insights into how different types of knowledge updates affect model behavior and stability.**

## D.2 KNOWLEDGE EDITING

Next, a big portion of recent literature has focused on understanding and modifying the internal knowledge of Large Language Models (LLMs), post-training. Such knowledge editing aims to alter specific facts or associations within the model without the need for full retraining.

Geva et al. (2020) were among the first to show that transformer Feed-Forward Network (FFN) layers act as unnormalized key-value stores encoding relational knowledge inside LLMs. This observation was later confirmed and complemented by others (?Dai et al., 2021) before being leveraged by subsequent work to master the editing of internal memories. ? introduced ROME (Rank-One Model Editing), a method that uses causal tracing to empirically locate the layers essential to encoding a given association. They then modify these modules by applying small rank-one changes. To identify the relevant modules, they run the network multiple times, introducing corruptions to the input sequence to disturb the inference, and then restore individual states from the original non-corrupted pass. But this work an others worked only on single edits, and were often evaluated one edit at a time, starting each time from a fresh pre-trained model. The same authors later developed MEMIT, which follows the same causal tracing principle but with the goal of scaling up to 10,000 edits in bulk(Meng et al., 2022b). Similarly, Dai et al. (2021) leveraged the identification of knowledge neurons to perform "knowledge surgery" – editing factual knowledge within Transformers without the need for additional fine-tuning. Zhu et al. (2020) approached the knowledge modification task as a constrained optimization problem. Their work found that constrained layer-wise fine-tuning emerges as an effective method for modifying the knowledge that Transformers learn, suggesting a different pathway for knowledge editing inside LLMs. De Cao et al. (2021) proposed KNOWLEDGEEDITOR, which achieved knowledge editing by training a hyper-network with constrained optimization to modify specific facts without fine-tuning or changing the overall stored knowledge. The method was demonstrated on smaller models like BERT for fact-checking and BART for question answering, achieving consistent changes in predictions across different formulations of queries.

Li et al. (2023) empirically investigate the pitfalls of knowledge editing in LLMs, revealing two critical issues: logical inconsistencies between multiple edits (like contradictory relationship updates) and knowledge distortion (where edits irreversibly damage the model's knowledge structure). Through carefully designed benchmarks CONFLICTEDIT and ROUNDEDIT, they demonstrate that current editing methods struggle with these challenges, particularly when handling reverse relationships or composite logical rules. While their work focuses on identifying limitations in maintaining logical consistency across edits, our paper takes a complementary cognitive-inspired perspective by addressing how models handle contradictions with their existing knowledge base.

Similarly, Huang et al. (2024) empirically investigate causes of performance degradation during knowledge editing in LLMs. They show degradation correlates with editing target complexity and L1-norm growth in edited layers. Their proposed Dump for Sequence (D4S) method regulates layer norm growth using O(1) space complexity, enabling multiple effective updates while minimizing model degradation. Their work provides valuable insights into the mechanisms of model degradation during knowledge editing.

Tan et al. (2023) propose MALMEN, a scalable hypernetwork approach for editing Large Language Models by aggregating parameter shifts using a least-squares formulation. While previous editing methods like MEND (Mitchell et al., 2022) could handle only a few facts simultaneously, MAL-MEN can efficiently edit thousands of facts while maintaining comparable performance. Their key innovation lies in separating the computation between the hypernetwork and LM, enabling arbitrary batch sizes and reducing memory requirements. Their empirical results show that MALMEN can edit hundreds of times more facts than MEND while maintaining similar performance levels, though they note that the method still struggles with generalizing to rephrasing not seen during training. Like other editing approaches, MALMEN focuses on the mechanics of (by design conflicting) updates.

**Unlike all the work above, our goal in this work is not to edit existing knowledge (which is conflicting by design according to our definition), but to understand the fundamental mechanisms and phenomena that govern how LLMs integrate new information with existing knowledge, contrasting contradictory and non-contradictory updates. By taking inspiration from humans' cognitive-dissonance, we reveal critical insights about the nature of knowledge representation and updating in these models.**

### D.3 RELATED EMPIRICAL AND THEORETICAL FINDINGS

To the best of our knowledge, we are the first to report the systematic catastrophic effect of contradictory updates on completely unrelated knowledge in LLMs (in addition to the benefits of differentiated plasticity in case of non-dissonant updates). A couple of contemporary findings are to an extent related to ours.

First, Hu et al. (2025) have recently shown that finetuning models to write insecure code leads to emergent misalignment across unrelated domains. Through the lens of our work, their findings may represent another manifestation of how contradictory updates propagate through neural networks. By training models to produce insecure code without disclosure (effectively contradicting their ethical alignment training) they observed widespread behavioral changes far beyond the coding domain, including anti-human viewpoints and deceptive behaviors. Similar to how our factual contradictions (e.g., changing nationality information) destroy unrelated factual knowledge, their ethical contradictions appear to disrupt the model's broader behavioral alignment. This parallel suggests that the catastrophic interference we document may be a more fundamental property of neural networks than previously recognized, affecting not only factual knowledge but potentially also learned ethical constraints.

Second, not related to LLMs but worth mentioning, Hiratani (2024) have analytically uncovered an interesting asymmetry in continual learning using a simple model as a playground for tractability (a linear teacher-student model with latent structure). They demonstrated analytically that this model struggles catastrophically when familiar inputs must be mapped to entirely new outputs (high input similarity, low output similarity), while performing relatively well in the opposite scenario (low input similarity, high output similarity). They further validated some key predictions using a single-hidden-layer network on a permuted MNIST task, but acknowledged that deeper networks might enable different adaptations to feature and readout similarity, calling for future work to adapt to more complex architectures and scenarios. Our empirical findings with various LLM models and facts reveal a potentially related phenomenon, *suggesting an asymmetry that extends beyond this simple model*. In fact, when we introduce contradictory updates (e.g. changing "Danielle Darrieux is French" to "English"), we observe catastrophic corruption of unrelated knowledge - conceptually similar to the high-input/low-output similarity scenario. Conversely, non-contradictory updates cause minimal interference. This parallel is particularly noteworthy given the vast differences in architecture complexity, domain, and scale between their controlled experiments and our work with LLMs. More research is needed to formalize this connection and determine under which conditions this represents a universal property of neural network learning more generally.

Finally, in a concurrent work, Hu et al. (2025) mathematically analyzed knowledge editing methods that use linear associative memory (like ROME and MEMIT) which directly modify specific weights rather than using gradient descent. They proved that these methods *will always inevitably suffer from interference* due to knowledge superposition in the model's parameter space. While their analysis is specific to these specialized editing techniques and primarily demonstrated with semantically related concepts (e.g., 'Vladimir Mayakovsky' and 'Vladimir Bukovsky' are highly superposed, meaning that editing one, will interfere with the other), our work reveals a parallel but distinct phenomenon in conventional gradient-based learning: contradictory updates cause catastrophic corruption of completely unrelated knowledge. This suggests that while the specific mechanisms differ between direct weight editing and gradient-based learning, both approaches encounter fundamental limitations when modifying existing knowledge. Our discovery of the stark asymmetry between contradictory and non-contradictory updates provides a complementary perspective to their superposition analysis, suggesting that the nature of the update itself is a critical factor in determining interference patterns across different training paradigms. A promising direction for future work would be to develop an equivalent "superposition" framework for gradient-based learning, potentially investigating whether

contradictory updates create more disruptive patterns in the model's representation space than non-contradictory ones.

# E    EXTRACTION OF HISTORICAL ACTIVATIONS AND GRADIENTS

We here detail our procedure for the extraction of activations and gradients. Source code is also available at `https://figshare.com/s/81f7108d823b5e08e8ec` for ultimate level of details and reproducibility purposes.

## E.1    PRELIMINARY NOTATION

We focus on the historical tracking of gradients of the outputs (grad_outs) and activations for four key matrices within each block of the transformer model: $\text{Attn}_{\text{c\_attn}}$, $\text{Attn}_{\text{c\_proj}}$, $\text{MLP}_{\text{c\_fc}}$, and $\text{MLP}_{\text{c\_proj}}$.

Given an input sequence $X \in \mathbb{R}^{B \times N \times d_{\text{model}}}$, where $B$ is the batch size, $N$ is the sequence length, and $d_{\text{model}}$ is the model dimension, the transformer block is defined as follows:

**Attention Layer:**    The attention mechanism computes query $Q$, key $K$, and value $V$ matrices:

$$Q = XW_Q, \quad K = XW_K, \quad V = XW_V$$

where $W_Q \in \mathbb{R}^{d_{\text{model}} \times d_{\text{key}}}$, $W_K \in \mathbb{R}^{d_{\text{model}} \times d_{\text{key}}}$, and $W_V \in \mathbb{R}^{d_{\text{model}} \times d_{\text{value}}}$ are trainable projection matrices.

The concatenated matrix $\text{Attn}_{\text{c\_attn}}$ is:

$$\text{Attn}_{\text{c\_attn}} = [Q, K, V] = XW_{\text{attn}}$$

where $W_{\text{attn}} = [W_Q, W_K, W_V] \in \mathbb{R}^{d_{\text{model}} \times (2d_{\text{key}} + d_{\text{value}})}$.

The attention context $\text{Attn}_{\text{context}}$ is computed as:

$$\text{Attn}_{\text{context}} = \text{softmax}\left(\frac{QK^T}{\sqrt{d_{\text{key}}}}\right)V$$

The projected attention output $\text{Attn}_{\text{c\_proj}}$ is:

$$\text{Attn}_{\text{c\_proj}} = \text{Attn}_{\text{context}}W_{\text{proj}}$$

where $W_{\text{proj}} \in \mathbb{R}^{d_{\text{value}} \times d_{\text{model}}}$.

**MLP Layer:**    The MLP layer consists of two linear transformations with an activation function $\sigma$:

$$\text{MLP}_{\text{c\_fc}} = \sigma(XW_{\text{fc}} + b_{\text{fc}})$$

where $W_{\text{fc}} \in \mathbb{R}^{d_{\text{model}} \times d_{\text{ff}}}$ and $b_{\text{fc}} \in \mathbb{R}^{d_{\text{ff}}}$.

The projected MLP output $\text{MLP}_{\text{c\_proj}}$ is:

$$\text{MLP}_{\text{c\_proj}} = \text{MLP}_{\text{c\_fc}}W_{\text{proj}} + b_{\text{proj}}$$

where $W_{\text{proj}} \in \mathbb{R}^{d_{\text{ff}} \times d_{\text{model}}}$ and $b_{\text{proj}} \in \mathbb{R}^{d_{\text{model}}}$.

## E.2    HISTORICAL GRADIENT AND ACTIVATION COLLECTION

Collecting a profile of neuron activity during training or simulation of training is needed as (i) input feature to know if a fact is dissonant, novel or known, and (ii) as means to identify where to locate targeted updates.

During training, we collect and cumulate the gradients of the outputs (grad_outs) and activations for the matrices $\text{Attn}_{\text{c\_attn}}$, $\text{Attn}_{\text{c\_proj}}$, $\text{MLP}_{\text{c\_fc}}$, and $\text{MLP}_{\text{c\_proj}}$. Let $t$ denote the training step. We collect activations at step $t$:

$$\text{Attn}_{\text{c\_attn}}(t), \text{Attn}_{\text{c\_proj}}(t), \text{MLP}_{\text{c\_fc}}(t), \text{MLP}_{\text{c\_proj}}(t)$$

as well as Gradient of the Outputs (grad_outs) at step $t$ :

$$\nabla L(\text{Attn}_{\text{c\_attn}}(t)), \nabla L(\text{Attn}_{\text{c\_proj}}(t)), \nabla L(\text{MLP}_{\text{c\_fc}}(t)), \nabla L(\text{MLP}_{\text{c\_proj}}(t))$$

In the remainder, we denote these, regardless of their provenance matrix, as:

$$A^l(t), G^l(t) \in \mathbb{R}^{B \times N \times d_{\text{out}}^l}$$

where $l$ denotes the layer, $B$ is the batch size, $N$ is the sequence length, and $d_{\text{out}}^l$ is the output dimension of layer $l$.

When needed, we standardize these metrics for each layer $l$ as follows:

$$\hat{A}^l(t) = \frac{A^l(t) - \mu_A^l(t)}{\sigma_A^l(t)}, \quad \hat{G}^l(t) = \frac{G^l(t) - \mu_G^l(t)}{\sigma_G^l(t)}$$

where $\mu$ and $\sigma$ are the mean and standard deviation computed over all dimensions of the respective tensor.

We then sum over the batch dimension:

$$S_{\hat{A}}^l(t)_{n,i} = \sum_{b=1}^{B} \hat{A}_{b,n,i}^l(t), \quad S_{\hat{G}}^l(t)_{n,i} = \sum_{b=1}^{B} \hat{G}_{b,n,i}^l(t)$$

Optionally[3], we can sum over the token dimension:

$$S_{\hat{A}}^l(t)_i = \sum_{n=1}^{N} S_{\hat{A}}^l(t)_{n,i}, \quad S_{\hat{G}}^l(t)_i = \sum_{n=1}^{N} S_{\hat{G}}^l(t)_{n,i}$$

The standardized and summed metrics are then accumulated across the training steps:

$$H\hat{A}_i^l = \sum_{t=1}^{T} S_{\hat{A}}^l(t)_i, \quad H\hat{G}_i^l = \sum_{t=1}^{T} S_{\hat{G}}^l(t)_i$$

where $T$ is the total number of training steps.

These historical activations $H\hat{A}^l$ and gradients $H\hat{G}^l$ provide cumulative measures of neuron activity over the training process. They help identify neurons that are heavily utilized (stubborn neurons) and those that are underutilized (plastic neurons), which is crucial for our targeted updates.

## F  DISSONANCE AWARENESS

### F.1  AUGMENTING THE COUNTERFACT DATASET WITH NOVEL FACTS

To generate unknown facts to augment the Counterfact dataset, we used GPT-3.5 with a prompt as follows:

```
1  Starting from this list of facts, can you create one data entry for each
      ↪ that concerns imaginary names and characters if necessary, while
      ↪ following the same logic.
2
3  For example, Danielle Darrieux's mother tongue is French => Becomes
      ↪ Machin De Machine's mother tongue is Kurdi (or Kinduli).
4
```

---

[3]We consider two approaches. In the first, we extract the activations and gradients corresponding to the last token (i.e., position $N$) in the sequence for each sample in the batch. This is reasonable since the last token is representative of the fact or information of interest in our datasets. In the second, we simply aggregate over all tokens, where we aggregate activations and gradients across all tokens in the sequence by computing statistical measures such as the mean or sum over the token dimension.

```
 5  Edwin of Northumbria's religious values strongly emphasize Christianity
        ↪ => Hamed Habib's religious values strongly emphasize Atheism (or
        ↪ Peace or..)
 6
 7  Try to make the old and new as far as possible from each other (e.g.,
        ↪ Kurdi is far from French, Kinduli is an imaginary language, etc.),
        ↪ while keeping some logic.
 8
 9  Write in JSON format, please (easy to parse):
10
11  - Danielle Darrieux's mother tongue is French
12  - Edwin of Northumbria's religious values strongly emphasize Christianity
13  - Toko Yasuda produces the most amazing music on the guitar
14  - One can get to Autonomous University of Madrid by navigating Spain
15  - Thomas Joannes Stieltjes was born in Dutch
16  - Anaal Nathrakh originated from Birmingham
```

**Example Generated Transformations:**

- Original: *"Toko Yasuda produces the most amazing music on the guitar."*
  Transformed: *"Zara Zorin produces the most amazing music on the theremin."*

- Original: *"One can get to Autonomous University of Madrid by navigating Spain."*
  Transformed: *"One can reach the Floating Academia of Zephyria by navigating through the Cloud Realms."*

- Original: *"Thomas Joannes Stieltjes was born in Dutch."*
  Transformed: *"Lorien Ilithar was born amidst the Elvish."*

These transformations help create novel facts unlikely to be known by the model, enabling us to evaluate its ability to handle unknown information effectively.

## F.2 ABLATION STUDY OF CLASSIFIER PERFORMANCE

We further report for the interested reader the results of an ablation study of the dissonance awareness classifier, evaluating its performance under different scenarios (fine-tuned vs. pre-trained models), feature sets (A, G, A+G), normalization strategies (None, Layer, Historical), and classifiers (Random Forests (RF) and Support Vector Machines (SVM)).

Table 9 presents a comprehensive set of classification results, including average accuracy and F1 scores (with standard deviations) across different settings. The best results for each classifier are denoted with a ⋆ and reported earlier in Table 2 in the main paper.

## F.3 EXPLANATION OF FEATURE IMPORTANCE

To further understand the discriminative power of different features, we analyzed the feature importance scores derived from the RF classifier.

First, as earlier mentioned in Fig.6 in the main paper, gradient-based features are substantially more important than activation-based features. This suggests that fine-tuning leads to more discriminative gradients, possibly due to the model overfitting on the known facts, resulting in near-zero gradients for known facts and higher gradients for novel or conflicting facts. In contrast, for the pre-trained model, both activation and gradient features contribute significantly, indicating that combining internal representations and learning dynamics is beneficial for classification.

Complementary to Fig.6, block importance reported in Fig. 7 reveals that, in the pre-trained model all transformer blocks tend to contribute relatively equally to the classification task, with the last layers contributing less. The finetuned model, on the other hand shows a slightly different tendency where the earlier layers contribute less. More work is clearly needed to understand such differences. This paper focuses only on feasibility of the entire cognitive-dissonance approach, leaving more elaborate evaluations for future work.

Table 9: *Ablation study of dissonance awareness:* Classification Results for Different Scenarios, Feature Sets, Normalization strategies and Classifier. Average (and std) accuracy and F1 scores. ⋆ denotes the best combination for each classifier

| Scenario | Features | Normalization | Classifier | Accuracy | F1 Score |
|---|---|---|---|---|---|
| Finetuned | A+G | Null | SVM | 0.994 (0.004) | 0.994 (0.004) |
| | | | RF⋆ | 0.988 (0.001) | 0.988 (0.001) |
| | | Layer | SVM | 0.995 (0.001) | 0.995 (0.001) |
| | | | RF | 0.982 (0.005) | 0.982 (0.004) |
| | | Historical | SVM⋆ | 0.995 (0.001) | 0.995 (0.001) |
| | | | RF | 0.978 (0.003) | 0.978 (0.003) |
| | G | Null | SVM | 0.917 (0.009) | 0.918 (0.009) |
| | | | RF | 0.905 (0.008) | 0.906 (0.008) |
| | | Layer | SVM | 0.920 (0.003) | 0.921 (0.003) |
| | | | RF | 0.895 (0.007) | 0.896 (0.007) |
| | | Historical | SVM | 0.897 (0.004) | 0.898 (0.004) |
| | | | RF | 0.868 (0.014) | 0.870 (0.014) |
| | A | Null | SVM | 0.796 (0.005) | 0.796 (0.007) |
| | | | RF | 0.747 (0.012) | 0.745 (0.016) |
| | | Layer | SVM | 0.783 (0.013) | 0.784 (0.012) |
| | | | RF | 0.722 (0.009) | 0.720 (0.007) |
| | | Historical | SVM | 0.781 (0.009) | 0.781 (0.010) |
| | | | RF | 0.721 (0.010) | 0.719 (0.008) |
| Pretrained | A+G | Null | SVM | 0.944 (0.006) | 0.944 (0.006) |
| | | | RF⋆ | 0.928 (0.012) | 0.929 (0.011) |
| | | Layer | SVM | 0.949 (0.006) | 0.949 (0.006) |
| | | | RF | 0.909 (0.014) | 0.910 (0.013) |
| | | Historical | SVM⋆ | 0.947 (0.004) | 0.948 (0.003) |
| | | | RF | 0.925 (0.006) | 0.925 (0.006) |
| | G | Null | SVM | 0.904 (0.006) | 0.904 (0.006) |
| | | | RF | 0.891 (0.010) | 0.892 (0.009) |
| | | Layer | SVM | 0.902 (0.008) | 0.902 (0.007) |
| | | | RF | 0.859 (0.013) | 0.861 (0.011) |
| | | Historical | SVM | 0.915 (0.007) | 0.916 (0.006) |
| | | | RF | 0.879 (0.017) | 0.879 (0.016) |
| | A | Null | SVM | 0.909 (0.006) | 0.909 (0.006) |
| | | | RF | 0.894 (0.009) | 0.895 (0.007) |
| | | Layer | SVM | 0.905 (0.012) | 0.905 (0.011) |
| | | | RF | 0.876 (0.004) | 0.877 (0.003) |
| | | Historical | SVM | 0.900 (0.008) | 0.900 (0.007) |
| | | | RF | 0.881 (0.006) | 0.882 (0.006) |

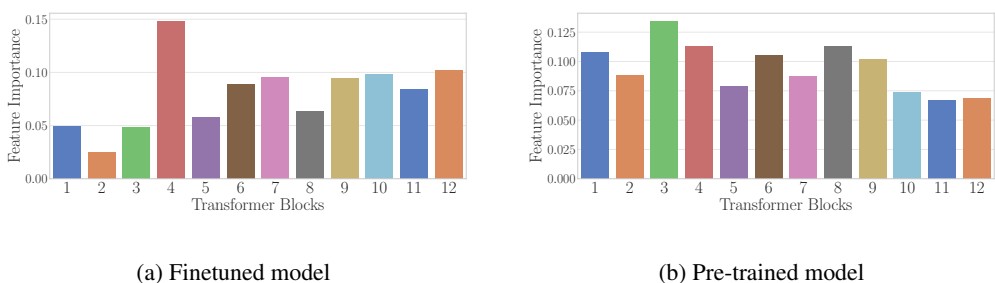

(a) Finetuned model          (b) Pre-trained model

Figure 7: *Block Importance.* Albeit differences are visible, the tendency is not as marked as for the activation vs gradient based feature importance in Fig.6 - GPT2-small

## F.4 LOCATION OF STUBBORN NEURONS

We also report the distribution of stubborn neurons across the transformer blocks in GPT-2 XL. Figures 8a and 8b show histograms of the number of stubborn neurons identified in each block for thresholds of 8,000 and 2,000 neurons, respectively.

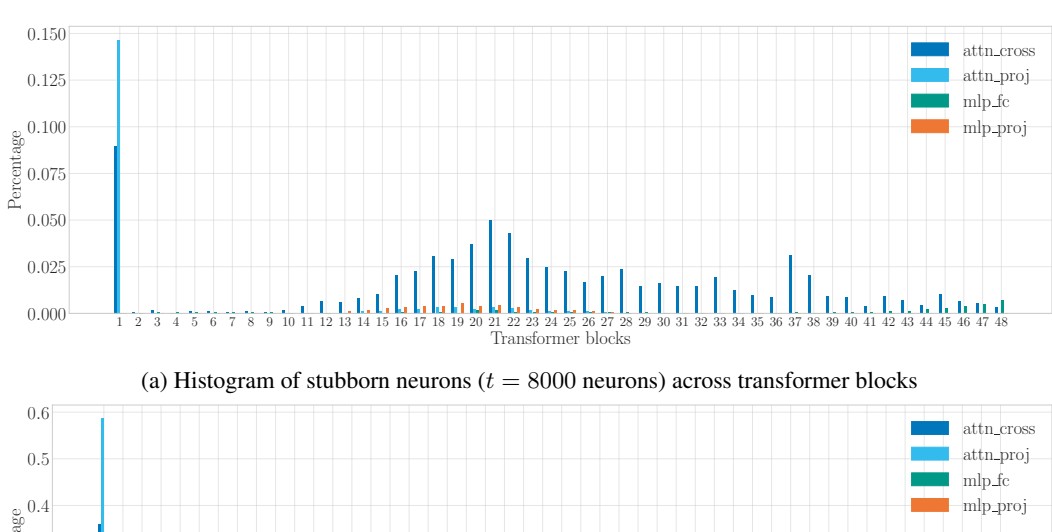

(a) Histogram of stubborn neurons ($t = 8000$ neurons) across transformer blocks

(b) Histogram of stubborn neurons ($t = 2000$ neurons) across transformer blocks

Figure 8: Distribution of stubborn neurons across GPT2-XL transformer blocks for different neuron thresholds to define stubbornness. (a) shows the distribution for $t = 8000$ neurons, while (b) corresponds to $t = 2000$ neurons.

Our analysis indicates that stubborn neurons are not uniformly distributed throughout the network. Instead, they curiousy tend to be concentrated in certain blocks, particularly in the first block and in certain middle layers of the transformer. This might suggest that these layers play a more significant role in encoding and retaining knowledge during training. Interestingly, $\text{Attn}_{\text{c\_attn}}$ concentrates much more of the stubborn neurons overall, with the exception of the first block where $\text{Attn}_{\text{c\_proj}}$ has a substantially higher share of stubborn neurons. The results are similar for both thresholds.

Overall, understanding the distribution of stubborn neurons can inform targeted update strategies by identifying which parts of the network are more critical for preserving existing knowledge.

### F.5 USING MODEL OUTPUT (INSTEAD OF INTERNAL STATE) AS FEATURES FOR DISSONANCE AWARENESS

In the main paper, we used activations and gradients as they were *readily available* in our experimental pipeline. We now further test whether using model output only, which is more easily available than internal gradients and activations can achieve similar performance on our scenario.

Each fact in our dataset is conceptually a statement involving a subject (s), relation (r), and object (o) (e.g., "Danielle Darrieux's mother tongue is French"). In this section, we extract features that capture increasing levels of detail about the model's predictions, related to what the actual facts are, leveraging both:

- Conditional probabilities $p(o|s, r)$ at different truncation points[4]
- Joint probability $p(s, r, o)$ of the full statement

---

[4]Since the object $o$ can span multiple tokens, we extract features from the last $N$ tokens of each fact (we pick three, since most answers fit within that limit). For each token position, we compute both the truncated prompt probability $p(o|s, r)$ by removing the token and subsequent tokens, and the full sentence probability $p(s, r, o)$. This multi-token analysis ensures we capture the model's predictions across the entire span of the answer.

In more details, we extract the following features, with increasing complexity.

**Basic Token Probabilities** ($Feat_1$): For each of the last $N$ tokens (representing the answer), we collect the probability of the actual next token given the truncated prompt. These simple scalar features capture the model's direct confidence in the correct continuation. This has a dimensionality of $N + 1$ ($N$ truncation points plus full statement, so 4 in our case.)

**Top-$k$ Predictions Analysis** ($Feat_2$): Here, for each position in the answer, we collect the values and normalized indices of top-$k$ most likely next tokens. This captures both confidence distribution and ranking patterns. Similarly to the above, we compute this for both truncated prompts and full statements. Here, the dimensionality is $(N + 1) \times 2k$ ($k$ values and $k$ normalized indices for each position). We pick k=100.

**Distribution Features** ($Feat_3$): Here, we analyze the complete probability distribution over the vocabulary. For each position in the answer sequence, we construct histograms of the probabilities with $n_{bins}$ bins (here 100), capturing the full spectrum of the model's prediction patterns. We augment these distributions with indicator vectors that highlight the positions of ground truth tokens (the true next tokens of the current truncated fact), providing additional context about the model's accuracy. This results in a feature vector of dimensionality $(N + 1) \times n_{bins}$.

**Combined Features** ($Concat$): Here, we simply concatenate $Feat_1$, $Feat_2$, and $Feat_3$.

Tab. 10 shows the results over our dataset. We observe *a similar great performance when using the model outputs, compared to Activations and Gradients*. Model output achieves even better performance in case of pre-trained models. This is inline with our earlier observation that activations (what we're using now) are more important than gradients in the case of pre-trained models. This result is encouraging for future work, where we plan to (i) build more challenging classification datasets (than the simple facts in CounterFact) and (ii) build standalone classifiers to speed up the training of LLMs, by avoiding training on conflicting data.

| Strategy (dim) | Pretrained Model | | Finetuned Model | |
|---|---|---|---|---|
| | Accuracy | F1-Score | Accuracy | F1-Score |
| Feat.1 (4) | 0.852 | 0.856 | 0.850 | 0.855 |
| Feat.2 (800) | 0.602 | 0.588 | 0.600 | 0.581 |
| Feat.3 (400) | 0.540 | 0.452 | 0.543 | 0.464 |
| Concat (1204) | 0.983 | 0.983 | 0.978 | 0.978 |
| (A+G) (240) | 0.947 | 0.948 | 0.995 | 0.995 |

Table 10: Using output-only features for dissonance-awareness can achieve similar good performance to using our readily available activations and gradients, and even better in the case of the pre-trained model.

## G NON-DISSONANT UPDATES

### G.1 SIMILARITIES WITH LOTTERY TICKET

To assess the hypothesis that certain subnetworks within the language model are more conducive to integrating new information—a notion earlier named the lottery ticket hypothesis (Frankle & Carbin, 2018)—we designed an experiment to confirm this effect.

We first trained a model on 10,000 disjoint facts (referred to as Facts H) and identified the most active candidate neurons during this process, which we term *Lottery Ticket Neurons*. These neurons should form a preferred subnetwork for representing Facts H. Next, we started from a *fresh model* and trained on a new set of novel facts (Facts A), which are different from H, restricting updates to three distinct groups of neurons:

1. **Lottery Ticket Neurons**: Neurons highly active during the initial training on Facts H.

2. **Non-Lottery Neurons**: Neurons underutilized during the initial training on Facts H.

3. **Random Neurons**: Neurons selected randomly from the entire network.

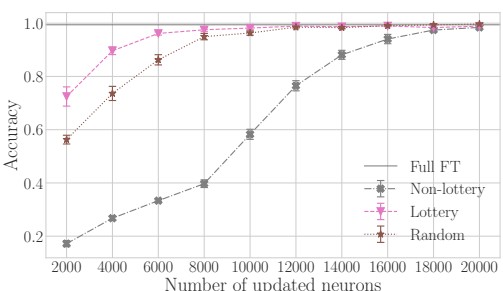

Figure 9: Lottery ticket

Figure 9 shows the accuracy of acquiring new knowledge when using each of these strategies, with the number of neurons varying from 2,000 to 20,000. Using the Lottery Ticket Neurons led to significantly better performance, reaching nearly 100% accuracy at 8,000 neurons, compared to around 40% for the Non-Lottery Neurons. The Random Neurons strategy also performed relatively well, interestingly suggesting that capturing even a few "anchor" neurons from the preferred subnetwork is sufficient to achieve good performance.

These results support the existence of preferred subnetworks within the model that are particularly effective for learning new information. Leveraging these subnetworks can enhance the efficiency of knowledge integration while preserving existing knowledge, an aspect that our candidate and specific strategies are already exploiting.

## G.2 HYPERPARAMETER SELECTION: LEARNING RATE AND BATCH SIZE FOR GPT2-XL

In our experiments, the first step is to conduct a hyperparameter search to determine the optimal learning rates and batch sizes for fine-tuning the model on our facts. Table 11 presents the performance of GPT2-XL on old and new knowledge across various learning rates and batch sizes. *Note that this optimal learning rate for full finetuning might turn out not enough for our targeted updates, since they use, by design, a smaller number of neurons.*

| Learning Rate | Batch Size | Epochs | Accuracy |
|---|---|---|---|
| 1e-06 | 64 | 5 | 0.271 |
| 1e-06 | 64 | 10 | 0.476 |
| 1e-06 | 64 | 20 | 0.694 |
| 1e-06 | 32 | 5 | 0.441 |
| 1e-06 | 32 | 10 | 0.641 |
| 1e-06 | 32 | 20 | 0.888 |
| 1e-06 | 16 | 5 | 0.582 |
| 1e-06 | 16 | 10 | 0.782 |
| 1e-06 | 16 | 20 | 0.984 |
| **1e-05** | **32** | **5** | **0.981** |
| 1e-05 | 32 | 7 | 0.997 |
| 1e-05 | 16 | 5 | 0.989 |
| 1e-05 | 16 | 7 | 0.997 |
| 1e-05 | 16 | 10 | 0.998 |
| 5e-06 | 32 | 5 | 0.853 |
| 5e-06 | 32 | 7 | 0.957 |
| 5e-06 | 32 | 10 | 0.996 |
| 5e-06 | 16 | 5 | 0.954 |
| 5e-06 | 16 | 7 | 0.996 |
| 5e-06 | 16 | 10 | 0.998 |

Table 11: Accuracy results for different learning rates, batch sizes, and epochs on 10k facts (GPT2-xl). We use the finetuning on 10k facts as a proxy to pick the hyperparameters of our later continual update experiments (learning rate, batch size and epochs). In bold, what we picked for GPT2-xl. Not shown here, for GPT2-small, we picked 5e-4.

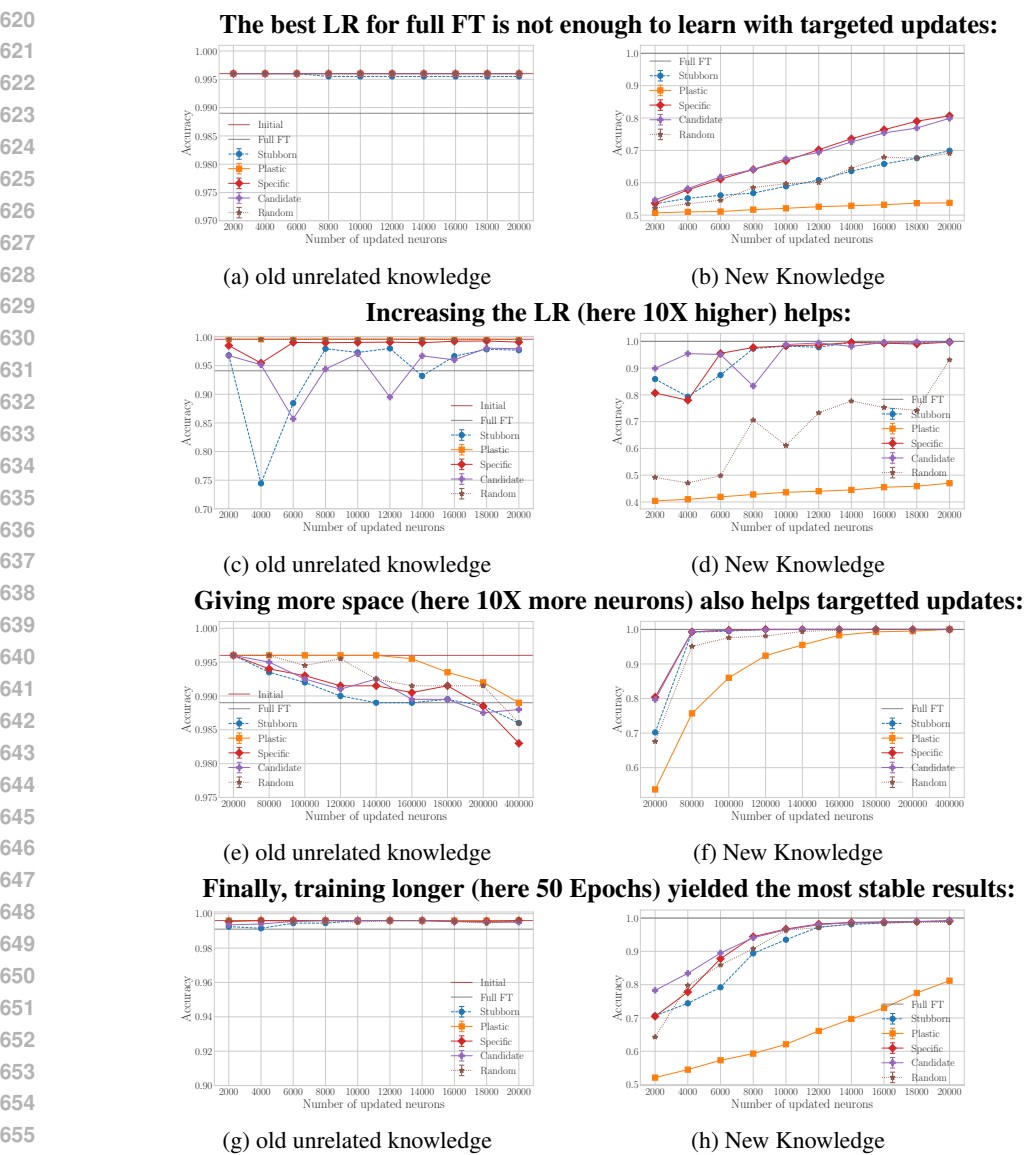

Figure 10: **Non-Dissonant** updates with **GPT2-XL** under various conditions. Overall the same trends as GPT2-small are confirmed: targeting stubborn neurons destroys old knowledge more and plastic neurons need more space or time to learn.

### G.3 COMPREHENSIVE ANALYSIS OF GPT2-XL NON-DISSONANT UPDATES

Figure 10 presents the accuracy of GPT-2 XL on old and new knowledge under various neuron update strategies and experimental conditions. We explored different configurations to understand how the model's larger capacity affects knowledge integration.

Our results reveal distinct scaling behaviors compared to GPT-2 small. With the optimal learning rate for GPT-2 XL (Figures 10a, 10b), we observe improved new knowledge acquisition while still preserving old knowledge. This means that although our carefully picked learning rate allows for efficient learning with full finetuning, learning with fewer neurons (as per our targetted strategies) seems harder than it was for GPT-2 small.

Increasing the learning rate by 10x (Figures 10c, 10d) or allocating 10x more neurons (Figures 10e, 10f) confirms that GPT-2 XL requires either higher learning rates or more extensive parameter updates compared to GPT-2 small to achieve effective learning with our targeted strategies.

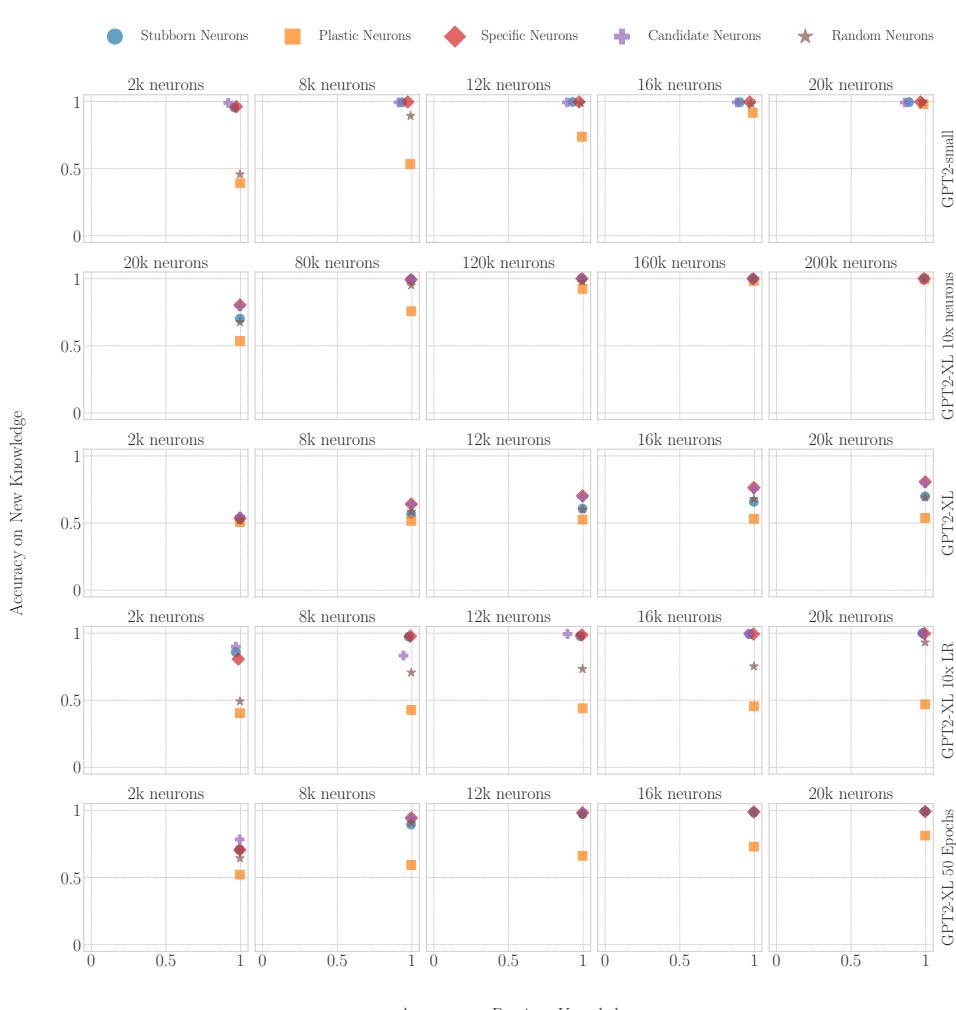

Figure 11: **Non-Dissonant** updates with GPT2-XL compared to small, a different visualization. Scatter plot of old (x) vs new (y) knowledge during non-dissonant updates. Same conditions as in Fig. 10. We can see clearly how in all cases, the accuracy on previous knowledge remains high. The lottery-ticket effect is also visible where free neurons struggle to efficient pack novel facts.

Similarly, extended training duration (50 epochs, Figures 10g, 10h) allows the model to better integrate new knowledge while preserving old information, indicating that longer training can also help overcome the limitations of sparse updates in larger models. Figure 11 summarizes these trade-offs across all configurations, highlighting how different hyperparameter choices affect the balance between preserving old knowledge and acquiring new information.

Finally, note that while GPT-2 XL's larger capacity naturally reduces interference with our tracked facts during non-dissonant updates, this improved performance is "deceptive" and should be interpreted cautiously: *we cannot measure potential effects on other pre-trained knowledge beyond our tracked facts*.

*These results highlight the methodological challenges in studying knowledge updates in larger models: their increased capacity can mask interference with tracked facts, making it harder to fully measure the impact of updates on the model's broader knowledge.* This underscores the importance of controlled experimental settings when studying fundamental properties of knowledge updating in neural networks.

# H DISSONANT UPDATES

## H.1 IMPACT OF NUMBER OF CONFLICTING FACTS

We examined the effect of varying the number of conflicting facts introduced during Dissonant updates. Figure 12 shows the performance metrics of GPT-2 small when editing 10, 100, and 1,000 facts, respectively.

Our findings show that as the number of conflicting facts increases, the impact on old unrelated knowledge retention becomes more pronounced, with all strategies experiencing significant degradation. The ability to learn new conflicting knowledge improves slightly with more facts, but overall performance remains suboptimal. The plastic and random neuron strategies tend to preserve old knowledge when editing a small number of facts (e.g., 10 facts), but their effectiveness diminishes as more conflicting information is introduced. Interestingly, the opposite effect is observed for new knowledge, where adding more facts seems to make it easier to learn new knowledge, for all strategies.

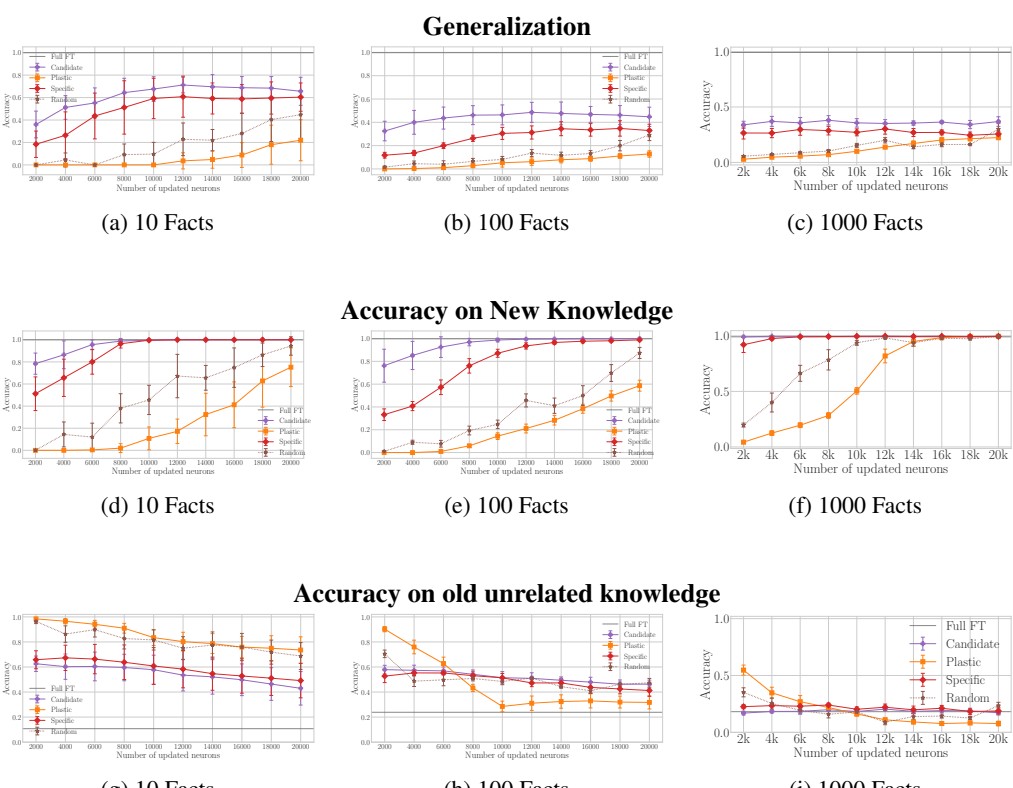

Figure 12: **Dissonant updates with GPT2-small - impact of the number of conflicting facts**. Each row represents a distinct metric: accuracy on the **Generalization** side dataset (paraphrased versions of the new facts), accuracy on **New Knowledge**, and Accuracy on **old unrelated knowledge**. Within each row, the subplots correspond to the number of conflicting facts introduced (**10 Facts**, **100 Facts**, and **1000 Facts**).

## H.2 COMPARATIVE PERFORMANCE OF EDITING METHODS

Our primary focus in this work is *not* on developing new model editing techniques. Most existing editing techniques focus on altering existing associations, and are hence by our definition dissonant by design. Our empirical findings in this work suggest another parallel path in which editing is abandoned in favor of non-dissonant variations where old knowledge is kept and contextualized.

However, to have an idea on how existing editing methods perform compared to our targeted strategies, we leverage `EasyEdit` (Wang et al., 2023) to benchmark two state-of-the-art model editing methods,

ROME (Meng et al., 2022a) and MEMIT (Meng et al., 2022b), under our same multi-fact experimental conditions .

Table 12 summarizes the performance of different strategies and editing methods. Some of our targeted update strategies obtain a higher harmonic mean compared to ROME and MEMIT. But the higher harmonic mean must not hide that the approaches are not directly comparable since they explore different regions of the pareto front, balancing new knowledge acquisition and old knowledge retention, as self-explained with colors and rankings in the table.

Table 12: Comparison of targeted neuron update strategies vs knowledge-editing literature, with a gradient from 0 (red) to 1 (green). Top-1,2 strategies annotated for all metrics and sample sizes.

| Samples | Strategy | Old (Unrelated) | New (Reliability) | Generalization | Harmonic Mean |
|---|---|---|---|---|---|
| | Full Finetune | 0.107 (0.082) | 1.000 (0.000) [1] | 0.576 (0.117) | 0.222 (0.116) |
| | MEMIT(Meng et al., 2022b) | 0.962 (0.079) [1] | 0.000 (0.000) | 0.000 (0.000) | 0.000 (0.000) |
| | ROME(?) | 0.891 (0.085) | 0.240 (0.182) | 0.180 (0.179) | 0.236 (0.235) |
| 10 | 8k Candidate | 0.596 (0.106) | 0.988 (0.024) [2] | 0.644 (0.128) [2] | 0.690 (0.058) [1] |
| | 20k Candidate | 0.430 (0.134) | 1.000 (0.000) [1] | 0.656 (0.125) [1] | 0.597 (0.116) |
| | 8k Specific | 0.638 (0.138) | 0.964 (0.039) | 0.512 (0.238) | 0.600 (0.183) |
| | 8k Stubborn | 0.622 (0.110) | 0.972 (0.030) | 0.544 (0.169) | 0.643 (0.103) [2] |
| | 8k Plastic | 0.909 (0.039) [2] | 0.020 (0.040) | 0.000 (0.000) | 0.000 (0.000) |
| | 8k Random | 0.827 (0.083) | 0.380 (0.132) | 0.092 (0.094) | 0.277 (0.098) |
| | Full Finetune | 0.238 (0.019) | 0.998 (0.003) [2] | 0.434 (0.089) | 0.398 (0.041) |
| | MEMIT(Meng et al., 2022b) | 0.976 (0.008) [1] | 0.004 (0.005) | 0.010 (0.007) | 0.003 (0.007) |
| | ROME(?) | 0.431 (0.108) | 0.300 (0.054) | 0.150 (0.036) | 0.240 (0.045) |
| 100 | 8k Candidate | 0.542 (0.035) [2] | 0.969 (0.033) | 0.462 (0.081) [1] | 0.591 (0.054) [1] |
| | 20k Candidate | 0.463 (0.032) | 0.999 (0.002) [1] | 0.447 (0.083) [2] | 0.552 (0.052) [2] |
| | 8k Specific | 0.531 (0.030) | 0.760 (0.063) | 0.263 (0.027) | 0.426 (0.024) |
| | 8k Stubborn | 0.530 (0.054) | 0.936 (0.048) | 0.398 (0.064) | 0.547 (0.063) |
| | 8k Plastic | 0.433 (0.029) | 0.059 (0.014) | 0.028 (0.017) | 0.052 (0.025) |
| | 8k Random | 0.508 (0.019) | 0.193 (0.038) | 0.065 (0.025) | 0.131 (0.039) |
| | Full Finetune | 0.182 (0.007) | 0.991 (0.009) | 0.442 (0.053) [1] | 0.341 (0.016) [2] |
| | MEMIT(Meng et al., 2022b) | 0.605 (0.107) [1] | 0.198 (0.053) | 0.100 (0.016) | 0.177 (0.028) |
| | ROME(?) | 0.152 (0.071) | 0.160 (0.093) | 0.067 (0.035) | 0.106 (0.058) |
| 1000 | 8k Candidate | 0.199 (0.014) | 0.996 (0.002) [1] | 0.380 (0.041) [2] | 0.345 (0.014) [1] |
| | 20k Candidate | 0.172 (0.018) | 0.996 (0.001) [1] | 0.369 (0.043) | 0.314 (0.028) |
| | 8k Specific | 0.240 (0.017) [2] | 0.993 (0.003) | 0.287 (0.039) | 0.345 (0.028) [1] |
| | 8k Stubborn | 0.200 (0.007) | 0.995 (0.001) [2] | 0.317 (0.024) | 0.327 (0.006) |
| | 8k Plastic | 0.218 (0.024) | 0.283 (0.026) | 0.070 (0.010) | 0.133 (0.013) |
| | 8k Random | 0.194 (0.026) | 0.663 (0.072) | 0.088 (0.008) | 0.165 (0.014) |

## H.3 MORE DETAILED FIGURES FOR SPECIFIC NUMBERS OF NEURONS

Tables 13, Figs. 14, and 15 provide detailed performance metrics for different neuron thresholds (20k, 8k, and 4k neurons, respectively) when editing 1,000, 100 and 10, conflicting facts using various strategies.

The results show that changing the number of neurons allocated for updates does not necessarily improve or degrade performance in the dissonant update scenario. In all cases, the model struggles to retain old unrelated knowledge while learning new conflicting information. The candidate and specific neuron strategies are consistently and significantly better than state of the art solutions, offering a slight advantage. However, they are still unable to effectively mitigate the destructive effects of dissonant updates, further motivating the neeed for both (i) dissonance awareness and (ii) proper conflict resolution.

## H.4 SCALING TO GPT2-XL

We extended our dissonant update experiments to GPT-2 XL to examine whether our observations about knowledge conflicts persist in larger models.

Figure 13 examines GPT2-XL's behavior when updating 1,000 conflicting facts using the optimal learning rate, as determined by our hyperparameter search. We compare three configurations: GPT-2 small (2,000 to 20,000 neurons) shown previously, GPT2-XL with the same range, and GPT2-XL

Table 13: Neuron Editing Results for N=20,000 Neurons

| Samples | Strategy | Accuracy A | Accuracy NOT(B) | Accuracy GEN | Harmonic Mean |
|---|---|---|---|---|---|
| | Full Finetune | 0.107 (0.082) | 1.000 (0.000) | 0.576 (0.117) | 0.222 (0.116) |
| | Specific | 0.491 (0.137) | 1.000 (0.000) | 0.604 (0.126) | 0.621 (0.109) |
| | Plastic | 0.735 (0.105) | 0.752 (0.175) | 0.220 (0.183) | 0.434 (0.185) |
| 10 | Stubborn | 0.449 (0.109) | 1.000 (0.000) | 0.616 (0.091) | 0.606 (0.084) |
| | Candidate | 0.430 (0.134) | 1.000 (0.000) | 0.656 (0.125) | 0.597 (0.116) |
| | Random | 0.688 (0.107) | 0.944 (0.083) | 0.448 (0.212) | 0.579 (0.222) |
| | Full Finetune | 0.238 (0.019) | 0.998 (0.003) | 0.434 (0.089) | 0.398 (0.041) |
| | Specific | 0.412 (0.046) | 0.988 (0.005) | 0.330 (0.054) | 0.460 (0.046) |
| | Plastic | 0.317 (0.052) | 0.586 (0.048) | 0.128 (0.028) | 0.233 (0.035) |
| 100 | Stubborn | 0.435 (0.043) | 0.999 (0.002) | 0.427 (0.085) | 0.528 (0.057) |
| | Candidate | 0.463 (0.032) | 0.999 (0.002) | 0.447 (0.083) | 0.552 (0.052) |
| | Random | 0.474 (0.035) | 0.874 (0.048) | 0.292 (0.048) | 0.444 (0.036) |
| | Full Finetune | 0.182 (0.007) | 0.991 (0.009) | 0.442 (0.053) | 0.341 (0.016) |
| | Specific | 0.188 (0.033) | 0.995 (0.002) | 0.257 (0.025) | 0.292 (0.035) |
| | Plastic | 0.077 (0.021) | 0.996 (0.002) | 0.224 (0.018) | 0.160 (0.027) |
| 1000 | Stubborn | 0.185 (0.010) | 0.992 (0.005) | 0.327 (0.013) | 0.317 (0.012) |
| | Candidate | 0.172 (0.018) | 0.996 (0.001) | 0.369 (0.043) | 0.314 (0.028) |
| | Random | 0.235 (0.029) | 0.995 (0.003) | 0.300 (0.053) | 0.347 (0.041) |

Table 14: Neuron Editing Results for N=8,000 Neurons

| Samples | Strategy | Accuracy A | Accuracy NOT(B) | Accuracy GEN | Harmonic Mean |
|---|---|---|---|---|---|
| | Full Finetune | 0.107 (0.082) | 1.000 (0.000) | 0.576 (0.117) | 0.222 (0.116) |
| | Specific | 0.638 (0.138) | 0.964 (0.039) | 0.512 (0.238) | 0.600 (0.183) |
| | Plastic | 0.909 (0.039) | 0.020 (0.040) | 0.000 (0.000) | 0.0 |
| 10 | Stubborn | 0.622 (0.110) | 0.972 (0.030) | 0.544 (0.169) | 0.643 (0.103) |
| | Candidate | 0.596 (0.106) | 0.988 (0.024) | 0.644 (0.128) | 0.690 (0.058) |
| | Random | 0.827 (0.083) | 0.380 (0.132) | 0.092 (0.094) | 0.277 (0.098) |
| | Full Finetune | 0.238 (0.019) | 0.998 (0.003) | 0.434 (0.089) | 0.398 (0.041) |
| | Specific | 0.531 (0.030) | 0.760 (0.063) | 0.263 (0.027) | 0.426 (0.024) |
| | Plastic | 0.433 (0.029) | 0.059 (0.014) | 0.028 (0.017) | 0.052 (0.025) |
| 100 | Stubborn | 0.530 (0.054) | 0.936 (0.048) | 0.398 (0.064) | 0.547 (0.063) |
| | Candidate | 0.542 (0.035) | 0.969 (0.033) | 0.462 (0.081) | 0.591 (0.054) |
| | Random | 0.508 (0.019) | 0.193 (0.038) | 0.065 (0.025) | 0.131 (0.039) |
| | Full Finetune | 0.182 (0.007) | 0.991 (0.009) | 0.442 (0.053) | 0.341 (0.016) |
| | Specific | 0.240 (0.017) | 0.993 (0.003) | 0.287 (0.039) | 0.345 (0.028) |
| | Plastic | 0.218 (0.024) | 0.283 (0.026) | 0.070 (0.010) | 0.133 (0.013) |
| 1000 | Stubborn | 0.200 (0.007) | 0.995 (0.001) | 0.317 (0.024) | 0.327 (0.006) |
| | Candidate | 0.199 (0.014) | 0.996 (0.002) | 0.380 (0.041) | 0.345 (0.014) |
| | Random | 0.159 (0.032) | 0.784 (0.091) | 0.102 (0.014) | 0.169 (0.010) |

Table 15: Neuron Editing Results for N=4,000 Neurons

| Samples | Strategy | Accuracy A | Accuracy NOT(B) | Accuracy GEN | Harmonic Mean |
|---|---|---|---|---|---|
| | Full Finetune | 0.107 (0.082) | 1.000 (0.000) | 0.576 (0.117) | 0.222 (0.116) |
| | Specific | 0.673 (0.101) | 0.656 (0.168) | 0.264 (0.208) | 0.385 (0.182) |
| 10 | Plastic | 0.965 (0.021) | 0.000 (0.000) | 0.000 (0.000) | 0.0 |
| | Stubborn | 0.635 (0.062) | 0.764 (0.087) | 0.352 (0.115) | 0.506 (0.101) |
| | Candidate | 0.603 (0.101) | 0.864 (0.126) | 0.512 (0.106) | 0.613 (0.065) |
| | Random | 0.863 (0.066) | 0.144 (0.113) | 0.044 (0.062) | 0.169 (0.050) |
| | Full Finetune | 0.238 (0.019) | 0.998 (0.003) | 0.434 (0.089) | 0.398 (0.041) |
| | Specific | 0.553 (0.023) | 0.408 (0.040) | 0.137 (0.022) | 0.258 (0.029) |
| 100 | Plastic | 0.760 (0.054) | 0.000 (0.000) | 0.003 (0.003) | 0.0 |
| | Stubborn | 0.565 (0.060) | 0.705 (0.143) | 0.303 (0.077) | 0.460 (0.092) |
| | Candidate | 0.573 (0.041) | 0.852 (0.124) | 0.400 (0.102) | 0.548 (0.093) |
| | Random | 0.487 (0.043) | 0.090 (0.018) | 0.045 (0.023) | 0.082 (0.030) |
| | Full Finetune | 0.182 (0.007) | 0.991 (0.009) | 0.442 (0.053) | 0.341 (0.016) |
| | Specific | 0.235 (0.008) | 0.976 (0.012) | 0.265 (0.041) | 0.329 (0.025) |
| 1000 | Plastic | 0.348 (0.049) | 0.125 (0.021) | 0.047 (0.006) | 0.093 (0.009) |
| | Stubborn | 0.203 (0.013) | 0.989 (0.006) | 0.315 (0.031) | 0.329 (0.016) |
| | Candidate | 0.184 (0.013) | 0.996 (0.001) | 0.370 (0.045) | 0.327 (0.025) |
| | Random | 0.254 (0.049) | 0.400 (0.085) | 0.072 (0.006) | 0.146 (0.010) |

with ten times more neurons (20,000 to 200,000). The latter was shown effective in packing new knowledge compared to (2000 to 20000) range in non-dissonant updates.

First, while GPT2-XL still requires more neurons than GPT-2 small to effectively learn new conflicting knowledge, as seen earlier, the key finding concerns unrelated knowledge retention: regardless of model size or neuron allocation, we observe significant degradation of old, unrelated knowledge across all strategies.

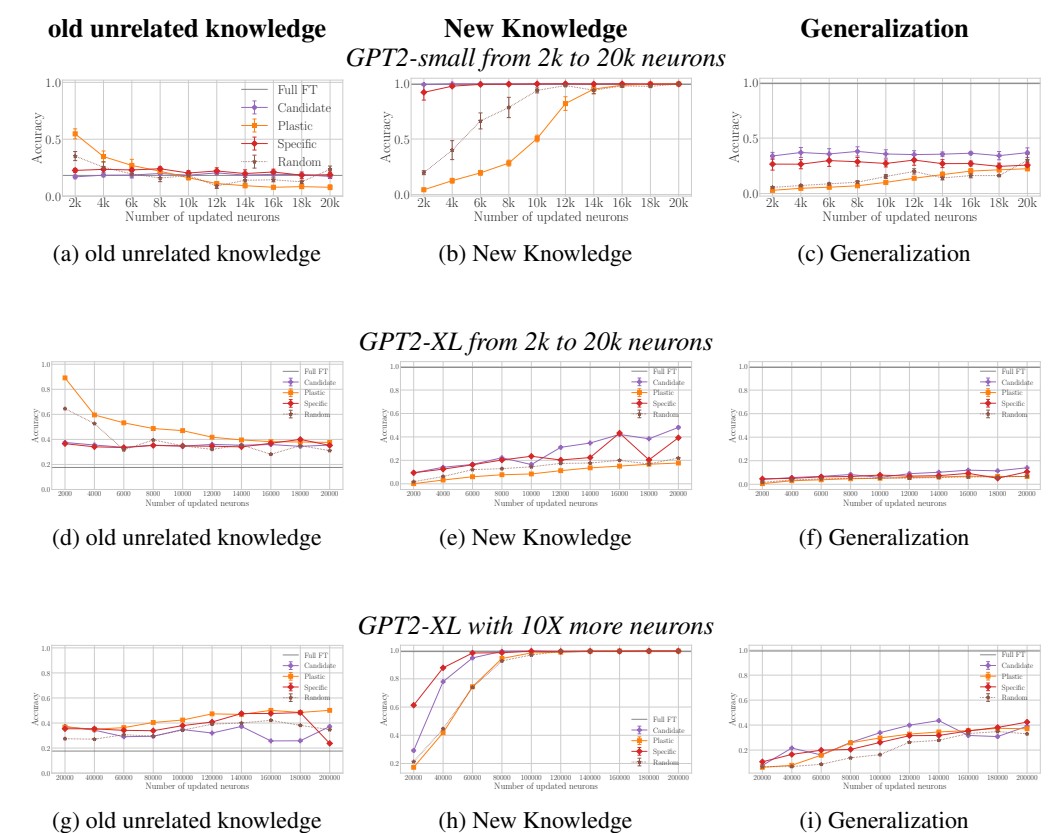

**old unrelated knowledge**     **New Knowledge**     **Generalization**

*GPT2-small from 2k to 20k neurons*

(a) old unrelated knowledge     (b) New Knowledge     (c) Generalization

*GPT2-XL from 2k to 20k neurons*

(d) old unrelated knowledge     (e) New Knowledge     (f) Generalization

*GPT2-XL with 10X more neurons*

(g) old unrelated knowledge     (h) New Knowledge     (i) Generalization

Figure 13: **Dissonant updates with GPT2-XL: whether the model learns new knowledge or not, old unrelated knowledge is severely destroyed regardless of the strategy** Experiments with 1000 facts using the best learning rate we found for Full Finetuning.

Interestingly, this degradation persists even when using fewer neurons and when the model fails to effectively learn the new conflicting information (2k to 20k). These results strongly suggest that the destructive impact of conflicting updates on existing knowledge is a fundamental property that remains present in larger models.

## I   LARGE LANGUAGE MODEL USAGE DISCLOSURE

In line with ICLR 2026 policies, we disclose the following LLM usage.

  (i) **Code development and debugging:** General purpose LLMs assisted with the training pipeline, visualization code, data generation, evaluation–prompt refinement, and plotting utilities. All generated code was reviewed, tested, and validated by the authors (web interface, without a coding assistant).

  (ii) **Writing assistance:** LLMs helped rewriting passages to attempt to enhance clarity and refine some technical descriptions. All scientific claims, hypotheses, interpretations, and conclusions are the authors' own.

  (iii) **Literature review and formulation:** LLMs helped discovered some related work, which served as seed to discover other related work; all references were independently verified by the authors.

The authors take full responsibility for all content, and LLMs served only as productivity tools (no contribution to core research ideas or discoveries).

