# OpenReview forum: "In Praise of Stubbornness: An Empirical Case for Cognitive-Dissonance Aware Continual Update of Knowledge in LLMs"
_ICLR.cc/2026/Conference — Submitted to ICLR 2026_

### Official Review · Reviewer_uQcp · 2025-10-29

**Soundness:** 2
**Presentation:** 1
**Contribution:** 2
**Rating:** 2
**Confidence:** 4

**Summary:**

This paper characterizes and investigates the problem of knowledge conflict under continual learning, demonstrating that language models trained on conflicting knowledge lead to damage in unrelated knowledge retention, which is moderate when trained on non-conflicting ones. Additional analysis reveals that the latter can be controlled by sparing frequently used neurens, but the former cannot be rescued by the same targeted update method.

**Strengths:**

S1: This paper characterizes the previously underrecognized and important phenomena of a model's collapse under continual learning with conflicting factual knowledge.

**Weaknesses:**

W1: The experimental setup is designed to control the scenario of non-dissonant and dissonant updates explicitly through the 'non-dissonant updates' phase; however, I believe this experimental protocol brings some (potentially confounding) factors that should be ruled out by additional experiments:
- The model's failure on phase 2 might be partially attributed to the parameter change during fine-tuning during phase 1, not originating from the knowledge conflict.
- Although the authors have acknowledged that a large portion of the knowledge injected during the first phase is already known to the model, some of them are not previously known (for example, ~40% of them for GPT-2-small), and newly added facts might behave slightly differently in the second phase compared to what is acquired throughout pretraining. For example, [1] showed that the two types of knowledge might behave differently during fine-tuning. It would be great to see an additional analysis on whether they show different behavior under knowledge conflict.



W2: Overclaimed connection to human cognition: It is true that human seems to perform belief revision in a significantly different manner compared to LLMs, but it is widely accepted in conceptual change [2] and belief revision [3] studies that humans do not simply append to update beliefs under dissonances, unlike the suggested "append-only updates that preserve both old and new knowledge with appropriate episodic context" in the discussion.



W3: There is a large room for improvement in presentation, and I believe the current quality of presentation is not good enough to recommend acceptance:
- Fig.1 is hard to parse because it contains too much information, which is not fully explained. For example, (1) What do 'candidate', 'specific', ..., 'finetuning' boxes in Fig.1 mean? I couldn't understand it until I saw the later sections (2) What does 'A&G' on the left side mean?
- L205-207 - How are the matrices $H$, $\hat G_n$, and $\hat A_n$ defined? If a ㅜnotation is used in the main text, then it should be defined in the main text, too.
- It is not provided what the X-axis values mean in Fig. 2.
- I could not understand the meaning of the table next to Fig.2, as the meanings and units of the numbers are not provided in the table and caption.
- The quantitative result for the experiment described in Sec 2.3 is displayed in Sec 5 without a pointer.
- Tab. 2 caption does not contain information on what is measured.




[1] https://arxiv.org/abs/2405.05904
[2] https://education.asu.edu/sites/g/files/litvpz656/files/lcl/chi_concpetualchangechapter_0.pdf
[3] https://plato.stanford.edu/entries/logic-belief-revision/

**Questions:**

Q1: When you employed LoRA as a control experiment for targeted editing approaches, how did you control the sparsity in LoRA updates (like rank) to ensure the fairness of the comparison with targeted training methods?

Q2: Is there any particular reason to display GPT-2-XL results in Fig. 2, despite stating "We therefore focus our main text on GPT-2-small results" in L188? Could you provide the same plot for GPT-2-small?

---

> ### Author Response · Authors · 2025-11-15
> **Response to Reviewer uQcp (1)**
>
> > W1: [...] I believe this experimental protocol brings some (potentially confounding) factors that should be ruled out by additional experiments: The model's failure on phase 2 might be partially attributed to the parameter change during fine-tuning during phase 1, not originating from the knowledge conflict.[...]
>
> This is a careful observation. We mentioned, perhaps not clearly enough, in the paper (line 166): "Finally, control experiments with a third round of non-dissonant updates, instead of dissonant, confirm the catastrophic effect is due to contradictions, not repeated updates".
>
> But even better, the experiments on GPT4 in appendix as well as our new experiments on Mistral and Llama remove any doubt since they are performed in a "single-shot" directly on the pre-trained model, without the initial phase of training on known facts. We observe similar patterns (the protocol here is i) start from a pretrained model then (ii) immediately training on dissonant (resp. non-dissonant facts) then finally (iii) evaluate on completely unrelated facts):
>
> - **llama3-BB instruct: retention of old knowledge**
> | Number of Updates | Dissonant | Non-dissonant |
> |-------------------|-----------|---------------|
> | 100               | 74%    | 98%        |
> | 1000              | 18%    | 95%        |
>
> - **Mistral-7B instruct: retention of old knowledge**
> | Number of Updates | Dissonant | Non-dissonant |
> |-------------------|-----------|---------------|
> | 100               | 63%       | 97%           |
> | 1000              | 20%       | 97%           |
>
> > W2: Overclaimed connection to human cognition: It is true that human seems to perform belief revision in a significantly different manner compared to LLMs, but it is widely accepted in conceptual change [2] and belief revision [3] studies that humans do not simply append to update beliefs under dissonances, unlike the suggested "append-only updates that preserve both old and new knowledge with appropriate episodic context" in the discussion.
>
> The reviewer is correct that conceptual change research shows humans revise their beliefs, but this operates at a different level than neural memory traces. The distinction is between behavioral output (which belief guides action) and memory traces (what information remains accessible). We are not arguing that humans behaviorally maintain both beliefs simultaneously in their decision-making (which would contradict belief revision research). Instead, even after conceptual change, humans retain episodic *memory traces* of prior beliefs (e.g. they remember what they believed before and when they changed their mind).
>
> This integration-vs-separation problem is fundamental in animal learning and memory (at neural/memory trace level): when encountering new information, does the system modify existing memories or create separate, context-dependent ones? Classic conditioning phenomena (extinction, renewal, spontaneous recovery, reconsolidation boundaries) demonstrate that animals maintain multiple representations rather than simply overwriting existing memories.
>
> One influential computational model addressing this is latent cause theory [1]. It posits that animals alternate between memory modification and memory creation based on prediction error: low prediction error leads to integration into existing representations, while high prediction error triggers creation of new latent causes.
>
> In any case, the connection to human cognition acted first solely as inspiration to guide the design of research questions: animals don't simply overwrite, they sometimes creates new disjoint memories, while keepig the old. And the first question was: what happens when LLMs are forced to overwrite when the information is contradictory with prior beliefs (something that animals totally avoid by creating new memories). The result is catastrophic. Even more interesting "append only updates" are useful, but only in case of incremental non-dissonant updates.
>
> This latter discovery ignited an ongoing neuroscience work where a model with "append-only updates that preserve both old and new knowledge with appropriate episodic context" successfully reproduces classical memory phenomena including extinction, renewal, and reconsolidation, providing the first neural-level mechanistic account aligned with latent cause theory.
>
> [1] The computational nature of memory modification. Elife, 6, e23763. (2017) https://elifesciences.org/articles/23763 .
>
> We believe our work is the first to make this connection, and the first to show these novel empirical results, with potential to open an alternative path for continual learning research.
>
> > W3: There is a large room for improvement in presentation
>
> We agree with most if not all presentation comments (many issues were due to a last minute restructuring, moving items to appendix). We can easily fix it, if this is the only remaining concern.

---

> > ### Author Response · Authors · 2025-11-15
> > **Response to Reviewer uQcp (1)**
> >
> > Questions:
> > > Q1: When you employed LoRA as a control experiment for targeted editing approaches, how did you control the sparsity in LoRA updates (like rank) to ensure the fairness of the comparison with targeted training methods?
> >
> > Our primary comparison is between dissonant and non-dissonant updates, as function of targeted update strategies, not between the update strategies and LoRA. LoRA and full finetuning were the focus of the first emprical result (Sec.3), and were put back there as a reminder/reference. That being said, we used a fixed (r=8, alpha=16) across experiments.
> >
> > > Q2: Is there any particular reason to display GPT-2-XL results in Fig. 2, despite stating "We therefore focus our main text on GPT-2-small results" in L188? Could you provide the same plot for GPT-2-small?
> >
> > Fig.2 uses GPT-2-XL for "pedagogical clarity": the larger model produces a smoother, more Gaussian-like distribution that better illustrates the 'overused' vs. 'underused' neurons concept. We can defintiely add the GPT-2-small equivalent for consistency. Figure2 was meant to be a qualitative illustration. Unfortunately, an unfortunate typo removed the pifont package from our LaTeX preamble, which caused the cross marks (✗) to render as '55' instead (\ding{55}). The corrected table should appear as follows. The entire plot plus figure were meant to be illustrative not quantitative:
> >
> > | Strategy  | Avoid stubborn | Follow gradient preference |
> > |-----------|----------------|----------------------------|
> > | Plastic   | ✓              | ✗                          |
> > | Stubborn  | ✗              | ✗                          |
> > | Candidate | ✗              | ✓                          |
> > | Specific  | ✓              | ✓                          |

---

> > > ### Comment · Reviewer_uQcp · 2025-11-26
> > >
> > > Thank you for the comments.
> > >
> > > > We mentioned, perhaps not clearly enough, in the paper (line 166): "Finally, control experiments with a third round of non-dissonant updates, instead of dissonant, confirm the catastrophic effect is due to contradictions, not repeated updates".
> > >
> > > This sentence alone in L166 does not contain useful information to convince the readers that the catastrophic effect is due to contradictions. The authors might have intended to point to some experimental results in the appendix, but I couldn't see any indication of which figure or table to refer to. Also, I cannot find the information for the experimental protocol and results for this control experiment. Moreover, I think this kind of statement should be placed in the results or discussion section, not in the section for describing the experimental pipeline.
> > >
> > > While the clarification and additional results provided by the authors address my concern regarding W1, I would recommend adding a clear explanation in the main text with appropriate pointers to deliver them.
> > >
> > >
> > > > One influential computational model addressing this is latent cause theory [1]. It posits that animals alternate between memory modification and memory creation based on prediction error: low prediction error leads to integration into existing representations, while high prediction error triggers creation of new latent causes.
> > >
> > > The main theme of the paper (which the authors mentioned to ground their view) seems to be modeling the trade-off between memory modification and memory creation by inferring latent causes. I don't think it is clear enough how the concepts and experimental results provided by the authors connect to inferring latent causes, or how maintaining memory in an append-only manner (as claimed by the authors) and the memory modification mechanism in the mentioned paper are connected. For example, how should the 'latent cause' be interpreted in terms of dissonant knowledge?
> > > Overall, even after seeing the authors' response, I believe the connection between what this work actually did and what the authors claimed is somewhat hand-wavy.
> > > Therefore, I would recommend significantly tempering the tone or providing a full discussion supported by a more thorough literature survey when discussing the connection to cognitive science.
> > >
> > > > We agree with most if not all presentation comments (many issues were due to a last minute restructuring, moving items to appendix). We can easily fix it, if this is the only remaining concern.
> > >
> > > Yes, I agree that presentation issues are usually fixable with minimal effort, but most of the points I provided are non-trivial (say, they are not mere typos) and need further clarification. As the authors mentioned that they can easily fix it, and ICLR does admit uploading revisions during the rebuttal period, I expect the authors to provide at least clarifications on each point or to upload a revision resolving the presentation issues. In the current status, I don't believe "many issues were due to a last-minute restructuring" and "we can easily fix it" resolves my concern regarding the presentation.
> > >
> > >
> > > Therefore, I will keep my initial score.

---

> > > > ### Author Response · Authors · 2025-12-04
> > > >
> > > > Thank you for your continued engagement with our work. We address your remaining concerns below.
> > > >
> > > > **W1 (Confounding factors):**
> > > >
> > > > > "While the clarification and additional results provided by the authors address my concern regarding W1, I would recommend adding a clear explanation in the main text with appropriate pointers to deliver them."
> > > >
> > > > We agree completely. Given uncertainty from other reviews about whether to invest further effort, we had hesitated to revise. We have now added the single-shot Llama/Mistral experiments and a clear explanation of the confounding factor controls to Appendix A.1.1, with pointers from Section 3. The paper now stands on its own without requiring the rebuttal for this evidence.
> > > >
> > > > **W2 (Connection to cognitive science):**
> > > >
> > > > We respectfully clarify that biological cognition served solely as *inspiration* for our research questions, not as a theoretical framework requiring precise mapping. The paper explicitly frames it this way throughout:
> > > >
> > > > - "This difference between human and artificial cognition **led us to investigate** a crucial question..."
> > > > - "Next, the brain **offers a second clue**..."
> > > > - "**This leads us to investigate** a second question..."
> > > > - "Our discoveries **inspire an intriguing hypothesis which deserves exploration in future work**..."
> > > > - "**While the exact nature of biological knowledge integration remains an active area of research**, our empirical findings motivate exploring..."
> > > >
> > > > Our empirical findings (that counterfactual updates cause catastrophic forgetting of unrelated knowledge while non-dissonant updates do not) stand independently of whether the cognitive inspiration is precisely correct.
> > > >
> > > > Just as birds inspired airplanes, or brains inspired neural networks, without claiming equialence, our cognitive framing motivated productive research questions without claiming theoretical equivalence. We hope the reviewer weighs the findings of the paper not its inspiration in the final judgment.
> > > >
> > > > Finally, the latent cause theory discussion in our rebuttal was offered as additional context for the inspiration, not as a claim requiring specification. We did not intend it to be evaluated as a theoretical contribution.
> > > >
> > > > **W3 (Presentation):**
> > > >
> > > > We have added one page of clarifications addressing the specific issues you raised, including proper explanation of Figure 1, definition of notation in the main text, axis labels, and table descriptions.
> > > >
> > > > We hope these revisions address your concerns and demonstrate our commitment to improving the manuscript.

---

### Official Review · Reviewer_7tn5 · 2025-10-29

**Soundness:** 2
**Presentation:** 2
**Contribution:** 1
**Rating:** 2
**Confidence:** 4

**Summary:**

This paper shows that fine-tuning language models on facts that contradict information seen before can cause forgetting of unrelated knowledge across the GPT-2 family, GPT-J, and GPT-4.1 models. It then shows that updating only the neurons with a high average historical gradient can preserve knowledge with non-dissonant updates, but not with dissonant updates. Finally, the paper shows that one may train an SVM on the activations and gradients to detect dissonance for the COUNTERFACT dataset reliably.

**Strengths:**

1. The impact of dissonant updates on unrelated knowledge is an interesting phenomenon.
2. The related work has a good overview of related works even as far back as a few decades.
3. The results on finetuning GPT-4.1 help establish that they hold even with newer models

**Weaknesses:**

1. The entire argument on selective plasticity and the connection to the human brain is very tenuous and anthropomorphizing. There is no rigorous analysis of why the learning of LLMs should be like the brain at all. There are much simpler explanations that must be investigated before these claims, such as that training an LM on contradicting facts teaches it to just output the opposite of what is true, similar to “emergent misalignment” [1].
2. The paper sets up the problem as the language models forgetting unrelated information when trained on contradicting facts—in this light it seems like the proposed method of updating only the top-N ranks of neurons based on historical gradient magnitude simply does not work for dissonant updates? (Figure 4)
3. The section on detecting dissonance:

    a. It is entirely conducted on the COUNTERFACT dataset which is synthetically constructed based on templates (not LLM-generated) and has examples which are very easily seen to be false. It is unclear if the results would hold on real-world examples where a statement may have previously been correct but is no longer, or more nuanced settings.

    b. Even if one detects dissonant updates, what then? Perhaps some of these updates are new, but correct information (we should probably not throw these away then)---while others are just misinformation. Besides, there are several documents that are not neatly categorized (consider code snippets, web scrapes, poorly formatted text, and so on); in this light I am not convinced that just training an SVM can help prevention of catastrophic interference.


[1] “Emergent Misalignment: Narrow finetuning can produce broadly misaligned LLMs,” Betley et. al., ICML 2025.

**Questions:**

1. Did you try the experiments on non-GPT models?
2. You provide some interesting hypotheses in the discussion section for the reasons why this behavior might be observed—did you try to validate these hypotheses?
3. Given your hypothesis that humans remember “X is used to be true, but now Y is true,” have you tried rewriting the facts Y in this form? I believe it would be a valuable contribution to demonstrate that this rewriting can help alleviate the issue of interference.

---

> ### Author Response · Authors · 2025-11-15
> **Response to Reviewer 7tn5 (1)**
>
> First, as mentioned to the other reviewers, given the current scores (6,2,2,2), we are realistic that acceptance is unlikely. We still believe the questions raised by this work can change the way we think about continual learning, so we share these comments both to clarify our position to future readers, and to gauge reviewer reactions.
>
> > The entire argument on selective plasticity and the connection to the human brain is very tenuous and anthropomorphizing [...] much simpler explanations [...] such as that training [...] teaches it to just output the opposite of what is true, similar to “emergent misalignment”.
>
> We do not think such explanations would be a weakness, we actually state them in our paper: (1) we discuss  the emergent misalignment paper in the extended related work and (2) explicitly emit the hypothesis that "learning to lie" is a potential explanation of the poor performance  (see our qualitative analysis in Appendix).  **Importantly, regardless of explanations, the empirical findings stand and are strong**: the asymmetry between dissonant and non-dissonant updates, the failure of all targeting strategies under contradiction.
>
> Second, we are *not* claiming that LLMs learn like brains or that neural mechanisms map directly to artificial networks.
> Rather, biological systems provided *productive heuristics* for designing experiments that led to discoveries we would not have made otherwise. Consider the discovery path:
>
> - *Initial observation from biology: Integration vs. Separation Problem*: When any learning system encounters new information, it faces a fundamental choice: modify existing knowledge (integration) or create separate memory traces (separation). Biological systems systematically avoid pure integration under contradiction (~high prediction error): classic phenomena like extinction and renewal in animal learning show that the original memory persists even after "unlearning" (e.g., fear returns in the original context). Latent cause theory [1] formalizes this: animals use prediction error to decide between integration (low error) vs. separation (high error, inferring a new "latent cause"). This inspired our question: what happens when LLMs are *forced* to integrate contradictory information, something biological systems systematically avoid all together?
>
> - *First empirical discovery*: Catastrophic interference from dissonant updates. This was unexpected and is independent of whether the brain analogy holds.
>
> - *Second biological heuristic*: Critical periods and selective plasticity in biological systems suggested investigating targeted updates. This led us to ask: does neuron selection affect dissonant vs. non-dissonant updates differently?
>
> - *Second empirical discovery*: The striking asymmetry (selective plasticity protects knowledge for non-dissonant updates but fails completely for dissonant ones). Again, this finding stands independently.
>
> - *Link back to biological connection*: This discovered asymmetry has now inspired an ongoing neuroscience work, where append-only architectures reproduced classical memory phenomena (extinction, renewal, reconsolidation boundaries), providing mechanistic accounts aligned with latent cause theory [1].
>
> [1] The computational nature of memory modification. Elife, 6, e23763. (2017) https://elifesciences.org/articles/23763 .
>
> **The biological inspiration functioned as a "guide" for experimental design, not as a claim of equivalence. The phenomena we discovered were previously unknown in the LLM literature and are our core empirical contributions, regardless of mechanistic interpretation**.
>
> > In this light it seems like the proposed method of updating only the top-N ranks of neurons based on historical gradient magnitude simply does not work for dissonant updates? (Figure 4)
>
> This is precisely one of our key findings: selective plasticity, a **mechanism that successfully protects knowledge during non-dissonant updates *fails* completely for dissonant ones**. This is **a discovery, not a methodological failure or limitation**.
>
> This asymmetry further led us to hypothesize that nature may have solved continual learning by combining two complementary protective strategies: (1) *avoiding* overwrites when surprise is high (integration vs. separation), and (2) *protecting* old neural structures for old knowledge (selective plasticity). This could be why selective plasticity works only when the update is incremental not dissonant. Continual learning research focused primarily on the second (plasticity-stability tradeoffs) while largely ignoring the first.
>
> > a. [...] It is unclear if the results would hold on real-world examples [...]
>
> Scaling this to more diverse datasets and domains is valuable and not trivial, deserving another work on its own (which is ongoing on our side).

---

> > ### Author Response · Authors · 2025-11-15
> > **Response to Reviewer 7tn5 (2)**
> >
> > > b. Even if one detects dissonant updates, what then? [...] in this light I am not convinced that just training an SVM can help prevention of catastrophic interference.
> >
> > We agree too. Our goal was never to use the SVM or RF in production. Our goal in this paper was instead to answer a scientific question: *Is there enough “signal” in the model’s internal or output statistics to distinguish (1) known, (2) novel non-contradictory, and (3) contradictory inputs?*
> >
> > This is not a trivial task even on the COUNTERFACT-derived setup. Our first attempts with simple closed-form proxies (for example surprisal, or weight-change magnitude alone) did not work reliably. The SVM/RF results show that a relatively low-capacity classifier can separate these three regimes with high accuracy, which we view as evidence that dissonance is in principle detectable, not as a finished prevention mechanism.
> >
> > > Did you try the experiments on non-GPT models?
> >
> > Yes, following a similar protocol of training on dissonant and non-dissonant updates and testing on completely unrelated factual knowledge. Experiments yielded similar results: the accuracy on totally unrelated previous knowledge drops when training on dissonant information compared to non-dissonant information.
> >
> > - **llama3-BB instruct: retention of old knowledge after training on dissonant vs non dissonant information**
> > | Number of Updates | Dissonant | Non-dissonant |
> > |-------------------|-----------|---------------|
> > | 100               | 74%    | 98%        |
> > | 1000              | 18%    | 95%        |
> >
> > - **Mistral-7B instruct: retention of old knowledge after training on dissonant vs non dissonant information**
> > | Number of Updates | Dissonant | Non-dissonant |
> > |-------------------|-----------|---------------|
> > | 100               | 63%       | 97%           |
> > | 1000              | 20%       | 97%           |
> >
> > > You provide some interesting hypotheses in the discussion section for the reasons why this behavior might be observed—did you try to validate these hypotheses? Given your hypothesis that humans remember “X is used to be true, but now Y is true,” have you tried rewriting the facts Y in this form? I believe it would be a valuable contribution to demonstrate that this rewriting can help alleviate the issue of interference.
> >
> > Thank you! We fully agree that contextualized rewrites of the form "X used to be true, now Y is true" are a very natural and valuable next step, and we are exploring them in an independent follow up work that builds on the present findings. In this paper, however, our focus is mainly diagnostic. We uncover three novel empirical results: (i) non dissonant updates are relatively safe, (ii) dissonant updates are catastrophically harmful even under selective plasticity, and (iii) dissonant inputs are in principle detectable from simple features. We believe this clear isolation and characterization of the contradiction induced failure mode is a substantial and self contained contribution, and a natural foundation on top of which more mitigation oriented studies, such as contextual rewrites (or iterative conflict resolution), can be built.

---

> > > ### Comment · Reviewer_7tn5 · 2025-11-23
> > > **Thank you for the response**
> > >
> > > Thank you for your response. I still feel that COUNTERFACT is a bit too artificial for us to have confidence that the findings can transfer to more realistic settings, and I personally am not that surprised that dissonant updates are more destructive than non-dissonant updates. Nonetheless, upon reading the rebuttal, I agree that it is still a useful thing to document and that it might enable interesting future work. In light of your response and the experiments on the Llama models, I am raising my score to 4.

---

> > > > ### Author Response · Authors · 2025-12-04
> > > >
> > > > Thank you for raising your score to 4 and for recognizing the value of documenting this phenomenon. We are grateful for your constructive engagement.
> > > >
> > > > As requested, we have incorporated the Llama3-8B and Mistral-7B experiments into the paper (Appendix A).
> > > >
> > > > We now also  extended to diverse dataset types beyond CounterFact (Appendix A.1.2), including misinformation and insecure code. The consistent pattern across model families and dataset types supports the generalizability of our findings.
> > > >
> > > > We hope these additions address your remaining concerns and that you would consider supporting acceptance of the paper.

---

### Official Review · Reviewer_4BJv · 2025-10-31

**Soundness:** 4
**Presentation:** 3
**Contribution:** 4
**Rating:** 6
**Confidence:** 4

**Summary:**

This paper presents an empirical investigation into how LLMs handle contradictory versus non-contradictory knowledge updates, drawing inspiration from human cognitive dissonance. The authors demonstrate that while LLMs can safely integrate non-contradictory information, attempting to update them with counterfactual information causes catastrophic corruption of unrelated knowledge. They also explore selective plasticity mechanisms and demonstrate that contradictory information can be detected with ≥95% accuracy using simple model features. The findings persist across multiple model scales.

**Strengths:**

1. This paper offers a hope for prevention of catastrophic interference so that it could have direct relevance to LLM deployment, particularly for models that undergo continual updates in production.
2. The method proposed in the paper for identifying stubborn and plastic neurons is straightforward and reasonable. Remarkably, the approach introduced in the paper aligns with the lottery ticket hypothesis in several respects and offers a novel possibility for discovering sparse subnetworks.

**Weaknesses:**

1.  The experiments are restricted to the GPT family (GPT-2, GPT-J, GPT-4.1). The generalizability of findings to other model architectures (e.g., LLaMA, Mistral, T5, BERT-style models) remains unclear. Different architectures may handle knowledge storage and updates differently, which could significantly affect the conclusions.
2. All experiments rely solely on CounterFact dataset. The findings may be specific to the types of factual knowledge and counterfactual patterns in this dataset.

**Questions:**

1. Can you provide evidence or theoretical justification for why these findings should generalize beyond GPT-family models?
2. How would you expect the findings to change with datasets with different domain-specific knowledge? Are there plans to validate on more diverse knowledge types?
3.  Can you provide concrete efficiency metrics for dissonance detection?

---

> ### Author Response · Authors · 2025-11-15
> **Response to Reviewer 4BJv**
>
> First, as mentioned to the other reviewers, given the current scores (6,2,2,2), we are realistic that acceptance is unlikely. We still believe the questions raised by this work are important for continual learning, so we share these comments both to clarify our position to future readers, and to gauge reviewer reactions.
>
> > Strengths: This paper offers a hope for prevention of catastrophic interference so that it could have direct relevance to LLM deployment, particularly for models that undergo continual updates in production. The method proposed in the paper for identifying stubborn and plastic neurons is straightforward and reasonable. Remarkably, the approach introduced in the paper aligns with the lottery ticket hypothesis in several respects and offers a novel possibility for discovering sparse subnetworks.
>
> Thank you for engaging deeply with the work: the lottery ticket hypothesis insights are indeed interesting and can be only noticed when engaging deeply with the paper. Thank you.
>
> > Weaknesses:
> > The experiments are restricted to the GPT family (GPT-2, GPT-J, GPT-4.1). The generalizability of findings to other model architectures (e.g., LLaMA, Mistral, T5, BERT-style models) remains unclear. Different architectures may handle knowledge storage and updates differently, which could significantly affect the conclusions.
>
> Since submission we have run additional experiments on LLaMA3 and Mistral instruction models, using a similar protocol (non-dissonant vs dissonant updates, evaluation on completely unrelated facts). The same qualitative pattern holds:
>
> - **llama3-BB instruct: accuracy on unrelated facts**
> | Number of Updates | Dissonant | Non-dissonant |
> |-------------------|-----------|---------------|
> | 100               | 74%    | 98%        |
> | 1000              | 18%    | 95%        |
>
> - **Mistral-7B instruct: accuracy on unrelated facts**
> | Number of Updates | Dissonant | Non-dissonant |
> |-------------------|-----------|---------------|
> | 100               | 63%       | 97%           |
> | 1000              | 20%       | 97%           |
>
> > All experiments rely solely on CounterFact dataset. The findings may be specific to the types of factual knowledge and counterfactual patterns in this dataset.
>
> In the extended related work we already point to signs that the phenomenon is more general than this specific dataset. For example:
>
> * Narrow finetuning on malicious code can turn a model broadly misaligned (e.g. wanting to harm humans).
> * Recent theory and experiments have shown that changing labels of existing images can be far more harmful than adding new labeled images, even in simpler classification models, *suggesting that what we observed could be a property of neural networks more generally and not only language models*.
>
> Finally, in a follow-up project we have started to test contradiction vs non-contradiction updates on other factual sources and domains.
>
> > Can you provide evidence or theoretical justification for why these findings should generalize beyond GPT-family models?
>
> As answered earlier, we empirically show it generalizes beyond the GPT family. Our appendix contains interesting pointers to the literature that hint to the fact that is a fundamental problem of neural networks more generally, not only LLMs.
>
> > How would you expect the findings to change with datasets with different domain-specific knowledge? Are there plans to validate on more diverse knowledge types?
>
> Thank you for the interesting ask. As mentioned earlier, this is part of our follow-up work, the same asymmetry appears.
>
> > Can you provide concrete efficiency metrics for dissonance detection?
>
> Our current dissonance detector is a feasibility study rather than a "final tool". The current detector is lightweight. The Random Forest classifier processes from ~100 to 1000 input features, and both training and inference costs are negligible compared to running the base model. The main cost lies in feature extraction. In the current setup, detection uses:
> - one forward pass to collect activations
> - one backward pass without parameter updates to collect gradients
> - simple aggregation operations (mean, std, quantiles per layer)
>
> So per candidate fact, the cost is roughly equivalent to one extra training-like step, on top of a normal forward pass. In Appendix F.5 we also experimented with using output-only features (no backward pass) reaching similar accuracy on our Counterfact-based dataset.
>
> In parallel, we're exploring simpler closed-form metrics for dissonance detection, with no success so far. For example, metrics based on magnitude of weight change alone fail to measure dissonance reliably. We are also working on this problem by building a more sophisticated dataset.

---

> > ### Author Response · Authors · 2025-12-04
> > **Additional experiments beyond counterfact**
> >
> > > All experiments rely solely on CounterFact dataset. The findings may be specific to the types of factual knowledge and counterfactual patterns in this dataset.
> >
> > We have now addressed also this concern by testing 5 new datasets with various degrees of "dissonance", from misinformation (conspiracy like statements likely present in training) to insecure code to common sense knowledge. The results consistently show a gradient in which the more dissonant the dataset, the larger the loss of unrelated knowledge.
> >
> > The new results are in Appendix A.2.

---

### Official Review · Reviewer_Bvbx · 2025-11-01

**Soundness:** 2
**Presentation:** 3
**Contribution:** 1
**Rating:** 2
**Confidence:** 5

**Summary:**

This paper investigates how Large Language Models (LLMs) handle new information, specifically contrasting the integration of facts that are consistent with existing knowledge ("non-dissonant") versus those that contradict it ("dissonant"). The authors find that dissonant updates cause catastrophic forgetting of unrelated knowledge, a phenomenon not observed with non-dissonant updates. They explore a "selective plasticity" approach, finding it protects knowledge during non-dissonant updates but fails for dissonant ones. Finally, they show that dissonant inputs can be detected with high accuracy.

**Strengths:**

1.   **Clear Problem Formulation:** The paper is well-motivated, using the intuitive analogy of cognitive dissonance to frame a critical problem in continual learning for LLMs. The distinction between dissonant and non-dissonant updates is a useful lens for analysis.
2.  **Rigorous Empirical Work:** The experiments are thoughtfully designed and clearly presented. The findings, particularly the stark contrast in how the model handles the two update types (Figure 3) and the asymmetry of selective plasticity (Figure 4), are compelling and well-illustrated.

**Weaknesses:**

However, the paper’s novelty and core contributions are severely undermined by its substantial overlap with the recent work of Sun et al. (ICLR 2025), “How new data permeates LLM knowledge and how to dilute it,” which the authors fail to cite or discuss. The framework introduced by Sun et al. provides a more general explanation for the phenomena observed here, suggesting that the effects described in this paper are a specific, less general instance of the “priming” and “surprisal” mechanisms identified in that work, thereby raising questions about the originality of the core findings.

1.  **Relation to the "Surprisal" Framework:** Sun et al. show that the degree to which new information undesirably "primes" unrelated knowledge is strongly predicted by its "surprisal" (i.e., the low probability of its keywords before training). The "dissonant" updates in this paper seem to be a specific instance of high-surprisal updates, while "non-dissonant" updates correspond to low-surprisal ones. The catastrophic effects observed with dissonant facts could therefore be an expected outcome of the general principle that high-surprisal data leads to larger, more disruptive updates. The analysis would be significantly strengthened by discussing this connection and clarifying what the binary "dissonance" framing offers beyond the continuous "surprisal" metric.

2.  **Parallels in Mitigation Strategies:** The proposed "selective plasticity" method, which spares high-gradient ("stubborn") neurons, has strong conceptual parallels with the "ignore-k" pruning method from Sun et al. The "ignore-k" method mitigates priming by removing the top-k percent of parameter updates, which are those with the highest gradient magnitudes. Both approaches are based on the same principle of isolating and either protecting or ignoring the largest gradients to prevent over-correction. The distinction between a neuron-level and a weight-level intervention would benefit from further justification to establish it as a methodologically distinct contribution.

3.  **Positioning of Contributions:** The paper's conclusion suggests "avoiding overwrites through episodic (temporal) contextualization" as a promising future direction. This idea is very similar to the "stepping-stone" text augmentation strategy already implemented and shown to be effective by Sun et al. This further underscores the need for this paper to situate itself within the context of this prior work to clearly delineate its novel contributions.

**Questions:**

Q1.   How might the findings on "dissonance" be framed in the context of the "surprisal" framework from Sun et al.? Does the binary distinction of contradiction offer unique insights that a continuous surprisal value might miss?

Q2. Could the authors elaborate on the key differences between their neuron-level "selective plasticity" and the weight-level "ignore-k" pruning from Sun et al.? What are the practical or theoretical advantages of the proposed approach?



**Suggestion**
* A discussion comparing this paper’s findings with those of Sun et al. would be invaluable for clarifying the specific advancements made here.
* It would be helpful if the authors could also perform a surprisal analysis and present quantitative results to illustrate the main differences between dissonant and non-dissonant fact updates.


If the authors can demonstrate that the phenomena they identify are not fully explained by the framework of Sun et al., I would be inclined to increase my score.

---

> ### Author Response · Authors · 2025-11-15
> **Response to reviewer Bvbx: differences with priming**
>
> First, despite believing in our work and its potential to  change the way we think about continual learning, given the current scores (6,2,2,2), we are considering withdrawal as a possible outcome. We are nonetheless sharing some early comments to leave a trace for future readers and to gauge reviewer reactions, based on which we will adjust our decision.
>
> Second, thank you for the reference to Sun et al and the comparative analysis. We notice that the review focused in its entirety on this paper, and so we will do.
>
> > However, the paper's novelty and core contributions are severely undermined by its substantial overlap with the recent work of Sun et al. (ICLR 2025), [...] thereby raising questions about the originality of the core findings.
>
> In short, both use similar stimuli but for different questions and to measure different side effects: Sun et al. use fictional or false snippets mainly to induce low keyword probability and then **they measure keyword priming : over‑use of that token under themed prompts** (e.g. models start using the term "vermillion" everywhere). We study explicit contradictions vs. non‑contradictions and **we measure retention of unrelated factual knowledge**. So while **their side effects are models that start using a particular word in inappropriate context**, **our side effect is a more severe behavioral change**: not just outputting a particular token, but starting to answer wrongly, or answering in different languages and so on (see qualitative analysis in Appendix). In short, their model learns to reuse unlikely words in unrelated contexts. In our case, the model becomes some sort of a stochastic "liar" *answering wrongly to "most" factual questions*. In a sense, this is a qualitatively different and more severe effect than keyword priming.
>
> > Relation to the "Surprisal" Framework: [...] clarifying what the binary "dissonance" framing offers beyond the continuous "surprisal" metric [...] Does the binary distinction of contradiction offer unique insights that a continuous surprisal value might miss?
>
> *Surprisal alone was not enough in our experiments:* surprisal was the first metric we tried for our classifier, but it did not work alone. Intuitively, unknown fictional facts are "surprising", yet not "catastrophic" (to unrelated prior knowledge) compared to contradictory ones (We have a classifier in Appendix based on output token probabilities which we can further discuss). Beyond surprisal, however, our current hypothesis is that contradictory updates cannot be reconciliated with existing knowledge, so perhaps the easiest path the gradients can find is a "shortcut": which is to "learn to lie", leading to poor performance on unrelated knowledge (even the amount of weight change between dissonant and non-dissonant updates is not enough to set them apart).
>
> > Parallels in Mitigation Strategies: The proposed "selective plasticity" method, which spares high-gradient ("stubborn") neurons, has strong conceptual parallels with the "ignore-k" pruning method from Sun et al. [...] would benefit from further justification to establish it as a methodologically distinct contribution.
>
> - *"Mitigation" strategies are fundamentally different: Sun's ignore-top-k method ** measures parameters with high gradients for the current knowledge being injected** (somewhat related to our "candidate" placement strategy, i.e., where would the gradients place the current update). **In contrast, when we talk about selective plasticity, we focus on protecting neurons that already store old knowledge**, what we call "stubborn neurons". Specifically, selective plasticity makes neurons storing prior knowledge resistant to change ("stubborn"), while less-used neurons remain plastic to accommodate new information.
>
> - *So-called "mitigation" results diverge in both papers:* While Sun et al.'s ignore-top-k method has a protective effect, our selective-plasticity approach fails under contradictions, even when we avoid all neurons that were important for past knowledge. Hence, the destructive effect is update-type-dependent, not pathway-dependent. In other words, it does not matter which neurons you protect or avoid: contradictory updates still produce widespread damage. Yet biological systems appear to use selective plasticity successfully. This puzzle motivated a new biological hypothesis that we are now investigating in a follow-up work, with implications that go beyond priming. We explain why in the next message.
>
> > The paper's conclusion suggests "avoiding overwrites through episodic (temporal) contextualization" as a promising future direction. This idea is very similar to the "stepping-stone" [...]
>
> It is indeed interesting and similar. The question with which we conclude remains however open: can it help protecting past unrelated knowledge, while successfully ingesting the new contextualized knowledge?

---

> > ### Author Response · Authors · 2025-11-15
> > **Research questions and implications of our findings**
> >
> > In reality, we don't have "mitigation" strategies in this work. The different placement strategies serve instead as empirical tools to probe how updates interact with stored knowledge. They were designed to answer biologically-inspired research questions, not to propose deployment-ready techniques.
> >
> > The broader motivation can find its roots in a classic problem in animal learning: every learning system must decide whether new information should be integrated into an existing memory or stored as a separate trace. Animals rely on prediction error (a relation between prior knowledge and new knowledge, which is different from simple token probability) and latent causes to make this decision[1]. When something strongly violates expectations, they typically create a new memory instead of overwriting the old one. This is why extinction never erases the original fear memory and why fear can return in the original context where fear acquisition happened. In simple terms, biological systems "never step in the same river twice" because every experience is contextually unique. They also seem to successfully use selective plasticity without forgetting prior knowledge.
> >
> > Our results reveal that LLMs behave differently. Even when we deliberately avoid the neurons that previously stored the old facts, contradictory updates still harm unrelated knowledge. This shows that the destructive effect is not due to how gradients flow through the network. It is because the model is forced to integrate incompatible information into the same parameters, something biological systems avoid altogether when prediction error is high, by allocating a new memory.
> >
> > This led us to conclude the work with a simple hypothesis: perhaps the brain's stability comes from two complementary mechanisms working together.
> > (i) Append-only updates that create new memories with their own episodic context instead of overwriting old ones.
> > (ii) Selective plasticity that protects old neural structures when learning incremental, non-contradictory information.
> >
> > Our results support this view: selective plasticity works well for non-dissonant updates, but once a contradiction appears, protecting specific neurons is not enough. This fits the idea that animals do not attempt to integrate contradictions into the same memory trace; they create a new one. This hypothesis is currently being explored in a neuroscience follow-up project with promising early results.
> >
> > Taken together, these findings point to something deeper than keyword priming. They reveal a severe limitation of gradient-based training and open a new perspective on the long-standing problem of continual learning: (i) avoid forcing the model to integrate contradictions, and (ii) preserve old neuron usage when learning new, non-dissonant information.
> >
> > [1] The computational nature of memory modification. Elife, 6, e23763. (2017) https://elifesciences.org/articles/23763

---

### Author Response · Authors · 2025-12-04

Dear AC,

Given the special circumstances, we structure our summary as follows: (1) what is now in the revised paper and (2) a clear timeline of reviewer concerns and our responses.

**Core contribution.** We found something that genuinely surprised us, and that could change how we think about continual learning : teaching an LLM counterfactuals destroys up to 80% of **totally unrelated** factual knowledge (with just 100 samples), while non-counterfactual updates largely preserve accuracy. Furthermore, selective plasticity (avoiding neurons that encode prior knowledge) fails completely for non-dissonant updates (succeeding for non-dissonant ones). Now validated across 6 models and 5 dataset types beyond CounterFact.

**What's Now in the Revised Paper (Main and Appendix):**
- **A.1.1**: Llama3-8B and Mistral-7B single-shot experiments (eliminates confounding from two-phase protocol)
- **A.1.2**: Five diverse datasets beyond CounterFact (misinformation, insecure code, CommonsenseQA, CSQA2, OpenBookQA)
- **Main text**: Presentation fixes (Split Figure 1 for clarity, Figure explanation, notation definitions from appendix to main, labels, table descriptions etc) citing new results in Appendix

---

### Reviewer-by-reviewer timeline

**Reviewer 4BJv (Score: 6→rebuttal→no answer )**
- *Concern*: Generalization beyond GPT family, CounterFact-only
- *Status*: **Fully addressed** Llama/Mistral and diverse datasets now in paper

---

**Reviewer 7tn5 (Score: 2 →rebuttal→4→ new results→no answer)** :
1. **Initial review (Score 2)**: "CounterFact is too artificial" "experiments on non-GPT models?"
2. **Our rebuttal**: Provided Llama3-8B and Mistral-7B experiments
3. **Reviewer response**: *"I am raising my score to 4"* but noted: *"I still feel that COUNTERFACT is a bit too artificial [..] to have confidence [...] transfer to more realistic settings"*
4. **Our follow-up**: Added **5 diverse datasets** (misinformation, insecure code, CommonsenseQA, CSQA2, OpenBookQA) showing consistent dissonance effects even with "softer" contradictions
-  The remaining concern was dataset diversity which we addressed it with 5 new datasets

---

**Reviewer uQcp (Score: 2→rebuttal→no answer)**
1. **Initial concerns**: W1 (confounding factors from two-phase protocol), W2 (overclaimed cognitive connection), W3 (presentation issues)
2. **Our rebuttal**: Provided single-shot Llama/Mistral experiments (addressing W1), clarified cognition as inspiration only (W2), fixed presentation (W3)
3. **Reviewer response** (verbatim): *"While the clarification and additional results provided by the authors **address my concern regarding W1**, I would recommend adding a clear explanation in the main text with appropriate pointers to deliver them."*
4. **Reviewer maintained score (2)** explicitly because: changes were in rebuttal, not in paper
5. **Our action**: We have **now added all changes to the paper** (Appendix A.1.1, presentation fixes in main text)
- The reviewer's own words state W1 is addressed. W3 (presentation) is fixed. W2 (cognitive connection) concerns supplementary rebuttal context, not our core claims. With changes now in the paper, the reviewer's stated conditions are met.

---

**Reviewer Bvbx (Score: 2→rebuttal→no answer)** (a single concern : overlap with Sun et al., AC judgment needed)
- *Our response*: Detailed technical differences:
  - Sun et al. measure **keyword priming** (models overuse unlikely tokens); we measure **destruction of unrelated factual knowledge**
  - Sun et al.'s "ignore-k" targets **current** update gradients; our selective plasticity targets **historically important** neurons
  - Their mitigation works; ours reveals that **no targeting strategy prevents damage from contradictions**
- In summary, our technical arguments demonstrate superficial similarities, we believe AC judgment is needed here (as per the discussions below)

---

### Why this paper merits acceptance

1. **Uncontested findings**: No reviewer disputes our core empirical results. All acknowledge the phenomenon is "interesting" (7tn5), the work is "rigorous" (Bvbx), and offers "hope for prevention of catastrophic interference" with "direct relevance to LLM deployment" (4BJv).

2. **All actionable concerns addressed**:
   - Generalization beyond GPT --> Llama, Mistral added
   - CounterFact too artificial --> 5 diverse datasets added
   - Confounding factors --> Single-shot experiments added
   - Presentation issues --> Fixed

3. **Remaining objections are about framing, not findings**: Whether our cognitive *inspiration* constitutes a rigorous theoretical framework was never our claim. Indeed, as clarified in our last answer to reviewer uQcp, the paper explicitly uses language like "led us to investigate," "offers a clue," and "deserves exploration in future work"

We respectfully ask the AC to weigh the strength of our **uncontested empirical contributions** (which reviewers acknowledged) against debates about how we frame our inspiration.

---

### Meta-Review · Area_Chair_BxFn · 2026-01-04

**Summary:**

The reviewers raised concerns primarily regarding the paper’s novelty and positioning with respect to closely related recent work (Bvbx), the generalizability of the findings beyond the GPT family and a single synthetic dataset (4BJv, 7tn5), the strength and clarity of the experimental protocol and its interpretation (uQcp), and substantial presentation and exposition issues that hinder comprehension and evaluation (uQcp). While several reviewers acknowledged that the documented phenomenon of catastrophic interference induced by contradictory updates may be interesting and potentially useful as a diagnostic observation, there was broad concern that the current manuscript does not yet clearly establish a sufficiently distinct, well-positioned, and rigorously supported contribution at the level expected for acceptance.

**Reviewer Concerns:**

Despite an extensive rebuttal and the addition of new experiments, several core concerns remain insufficiently resolved. Although the authors provided new results on LLaMA and Mistral models, alleviating concerns about restriction to the GPT family, reliance on COUNTERFACT and similarly controlled settings continues to raise doubts about the transferability of the findings to more realistic or diverse update scenarios (4BJv, 7tn5). In addition, while explanations were offered to address potential experimental confounds and to motivate the cognitive and biological framing, at least one reviewer found that key clarifications were not clearly incorporated into the main text and that the connection to cognitive theories remained insufficiently grounded in the presented experiments (uQcp). Finally, multiple non-trivial presentation and clarity issues, beyond minor typos, remain unresolved, limiting the paper’s accessibility and the reviewers’ ability to fully assess the claims (uQcp).

**Reviewer Scores:**

One reviewer (7tn5) explicitly increased their score to 4 following the rebuttal, citing the additional experiments and clarifications. The other reviewers did not indicate score changes and, based on their final comments, would likely maintain their original evaluations. Although the rebuttal addressed several empirical and scope-related concerns, especially regarding generalization beyond GPT models, the remaining issues related to novelty positioning, conceptual grounding, and presentation quality suggest that any further score increases would be limited and insufficient to change the overall outcome.

---

### Decision · Program_Chairs · 2026-01-26

Reject